## 1 Soil microbial necromass shapes global carbon stocks in agricultural and natural

### 2 ecosystems

- 3 Jing-li Lu<sup>a</sup>, Thomas W. Crowther<sup>b</sup>, Manuel Delgado-Baquerizo<sup>c</sup>, Wenjie Liu<sup>a</sup>, Yamin
- 4 Jiang<sup>a</sup>, Hongyang Sun<sup>d</sup>, Zhiqiang Wang<sup>d, \*</sup>
- 5
- <sup>6</sup> <sup>a</sup> Center for Eco-Environment Restoration Engineering of Hainan Province, School of
- 7 Ecology, Hainan University, Haikou, 570228, People's Republic of China;
- 8 <sup>b</sup> Institute of Integrative Biology, Department of Environmental Systems Science,
- 9 ETH Zürich, 8092 Zürich, Switzerland;
- 10 <sup>c</sup> Laboratorio de Biodiversidad y Funcionamiento Ecosistémico, Instituto de Recursos
- 11 Naturales y Agrobiología de Sevilla (IRNAS), CSIC, Seville, 41013, Spain;
- 12 <sup>d</sup> Sichuan Zoige Alpine Wetland Ecosystem National Observation and Research
- 13 Station, College of Grassland Resources, Southwest Minzu University, Chengdu,
- 14 610041, People's Republic of China
- 15
- 16
- 17 \* Corresponding author: Zhiqiang Wang
- 18 E-mail address: wangzq@swun.edu.cn

### 20 Abstract

Soil carbon (C) plays an essential role in regulating global C cycle and climate. 21 22 Microbial necromass is an important component of soil C, and yet the relative contribution of microbial necromass in shaping the global C stocks in agricultural and 23 24 natural ecosystems worldwide remains virtually unknown. In this study, we compiled data on fungal and bacterial necromass along with soil organic carbon (SOC) from the 25 26 0-20 cm soil layer across 486 study sites (145 agricultural and 341 natural ecosystems) to evaluate the relative contribution of fungal necromass C (FNC) and bacterial 27 necromass C (BNC) to SOC and the FNC/BNC ratio, after accounting for other biotic 28 and abiotic factors. Our results indicated that, in both agricultural and natural 29 ecosystems, the contribution of FNC to SOC significantly exceeded that of BNC, with 30 31 FNC contributing approximately twice as much as BNC to SOC. However, the contributions of FNC and BNC to SOC were markedly higher in agricultural 32 ecosystems than those in natural ecosystems, with a contrasting trend in the 33 FNC/BNC ratio. Soil physicochemical properties (C/N and clay) were the most 34 35 important predictors of the contributions of FNC and BNC to SOC in both ecosystems, while geographical factor (elevation) was the most important predictor of the 36 FNC/BNC ratio. Our study enhances the current level of understanding regarding 37 microbially mediated biogeochemical cycling and SOC dynamics, underscoring the 38 39 critical role of microbial necromass in the global C cycle.

40

41 Keywords: agricultural ecosystems, bacterial necromass carbon, fungal necromass

42 carbon, microbial necromass carbon, natural ecosystems

43

#### 44 1 Introduction

As the largest carbon (C) pool in the terrestrial biosphere, soil organic carbon (SOC) 45 46 plays a pivotal role in shaping the global C cycle and climate system (Bellamy et al., 2005; Crowther et al., 2015). The traditional paradigm is that SOC storage is 47 48 primarily formed directly based on plant material (Zhu and Miller, 2003; von Lützow 49 et al., 2008). However, emerging evidence demonstrates that the stable SOC pool is 50 predominantly composed of microbially derived organic compounds rather than 51 plant-derived residues, indicating that microbial necromass may in fact be a primary source of stable SOC (Kallenbach et al., 2016; Liang et al., 2017). Microbes generate 52 53 biomass by degrading plant-derived C, which is subsequently stabilized as microbial necromass through interactions with minerals and various processes within soil 54 aggregates (Liang et al., 2017). Although the living soil microbial biomass constitutes 55 only about 2% of SOC, microbial necromass carbon (MNC) can contribute up to 56 57 50%-80% of SOC (Liang and Balser, 2011; Kallenbach et al., 2016; Liang et al., 2019). As such, there is growing scientific attention on the forces driving the 58 accumulation of MNC and its contribution to SOC (Liang et al., 2017; Ni et al., 2020; 59 Luo et al., 2022; Zhou et al., 2023). In order to gain a comprehensive and in-depth 60 understanding of the role of MNC in the global C cycle, many studies have focused 61 on the relative contributions of fungal (FNC) versus bacterial necromass (BNC; 62 Zhang et al., 2021; Zhou et al., 2023; Xu et al., 2024). 63

The distinct roles of fungi and bacteria in stabilizing and decomposing SOC, 64 65 enable their necromass C to elucidate the kinetics of SOC storage and decomposition (Malik et al., 2016). To be more specific, the chemical composition and functional 66 characteristics of fungi and bacteria differ considerably, which may also contribute to 67 68 different mechanisms of SOC formation. The cell walls of fungi primarily consist of chitin (a nitrogen-containing polysaccharide) and β-glucans, whereas bacterial cell 69 70 walls are mainly composed of peptidoglycan-a complex of sugars and amino acids 71 (Lenardon et al., 2007). The intricate structures of chitin and  $\beta$ -glucans in fungal 72 necromass make them more resistant to decomposition compared to the typically more degradable bacterial necromass, leading to differences in their C decomposition 73 rates in soil (Xu et al., 2022). Wang et al. (2021a) suggested that the higher 74 75 contribution of FNC to SOC (exceeding 65%) than BNC (32-36%) may be not only due to the slower decomposition rates of fungal cellular compounds but also 76 attributable to the higher living fungal biomass relative to bacterial biomass in 77 terrestrial ecosystems. Previous studies have also indicated that the contributions of 78 79 FNC and BNC to SOC depended on the type of ecosystems (Wang et al., 2021a; Cao et al., 2023; Xu et al., 2024), and they were mainly affected by climatic and soil 80 physicochemical factors (e.g., soil pH and clay content; He et al., 2020; Zhang et al., 81 2023). However, these studies on the relative contributions of fungal necromass 82 83 carbon (FNC) and BNC to SOC, and their ratio (FNC/BNC), have predominantly focused on ecosystems categorized based on biotic communities and vegetation types, 84 such as forests (Chen et al., 2020), with limited attention given to ecosystems 85

86 classified by levels of human interference.

Terrestrial ecosystems can be broadly categorized into managed (agricultural) and 87 natural ecosystems (Hobbs et al., 2011; Keith et al., 2022). The agricultural 88 ecosystems, heavily reliant on human management, typically exhibit uniform plant 89 90 litter influenced by agricultural activities (Bohan et al., 2013). In contrast, natural ecosystems, which are self-sustaining, display greater diversity in litter and root 91 92 deposits, encompassing various plants and biological remains (Wu et al., 2019). 93 Minimal human interference enhances fungal mycelial networks and stable soil 94 aggregates, leading to higher FNC sequestration and contributions to SOC (Sanaullah et al., 2020; Sae-Tun et al., 2022). As key decomposers, fungi can break down 95 cellulose and other complex organic compounds (Hättenschwiler et al., 2005). Choi et 96 97 al. (2018) indicated that soil cellulose-degrading genes are associated with fungal activity and abundance. This suggests cellulose-rich soils may support higher fungal 98 diversity and abundance. The low cellulose and chitin abundance in agricultural 99 ecosystems might result from plant monoculture, while diverse plant inputs in natural 100 101 ecosystems lead to rich soil cellulose content and higher fungal abundance and FNC (Song et al., 2022). These findings suggest that FNC and BNC accumulation respond 102 differently to human interference and ecosystem characteristics, causing disparities in 103 the FNC/BNC ratio across ecosystems. Therefore, understanding global patterns and 104 105 drivers of FNC, BNC, and the FNC/BNC ratio in agricultural and natural ecosystems is crucial amid intensifying human activities and global changes. 106

107 In order to explore the global patterns and drivers of FNC, BNC and the

| study sites worldwide. The aims of this study were: (1) to quantify the contributions<br>of FNC and BNC to SOC and the FNC/BNC ratio in agricultural and natural<br>ecosystems; and (2) to investigate the primary driving factors influencing the<br>contributions of FNC and BNC to SOC and the FNC/BNC ratio, and determine<br>whether the influencing factors were consistent across these two ecosystem types. | 108 | FNC/BNC ratio in agricultural and natural ecosystems, we compiled data from 486       |
|---------------------------------------------------------------------------------------------------------------------------------------------------------------------------------------------------------------------------------------------------------------------------------------------------------------------------------------------------------------------------------------------------------------------|-----|---------------------------------------------------------------------------------------|
| of FNC and BNC to SOC and the FNC/BNC ratio in agricultural and natural<br>ecosystems; and (2) to investigate the primary driving factors influencing the<br>contributions of FNC and BNC to SOC and the FNC/BNC ratio, and determine<br>whether the influencing factors were consistent across these two ecosystem types.                                                                                          | 109 | study sites worldwide. The aims of this study were: (1) to quantify the contributions |
| ecosystems; and (2) to investigate the primary driving factors influencing the<br>contributions of FNC and BNC to SOC and the FNC/BNC ratio, and determine<br>whether the influencing factors were consistent across these two ecosystem types.                                                                                                                                                                     | 110 | of FNC and BNC to SOC and the FNC/BNC ratio in agricultural and natural               |
| <ul><li>112 contributions of FNC and BNC to SOC and the FNC/BNC ratio, and determine</li><li>113 whether the influencing factors were consistent across these two ecosystem types.</li></ul>                                                                                                                                                                                                                        | 111 | ecosystems; and (2) to investigate the primary driving factors influencing the        |
| 113 whether the influencing factors were consistent across these two ecosystem types.                                                                                                                                                                                                                                                                                                                               | 112 | contributions of FNC and BNC to SOC and the FNC/BNC ratio, and determine              |
|                                                                                                                                                                                                                                                                                                                                                                                                                     | 113 | whether the influencing factors were consistent across these two ecosystem types.     |

114

#### 115 2 Materials and methods

#### 116 2.1 Data collection

To clarify the contributions of FNC and BNC to SOC and the FNC/BNC ratio at 117 global scale, we compiled a comprehensive dataset from a range of peer-reviewed 118 119 papers published before 2022 through the Web of Science (http://apps.webofknowledge.com), Google Scholar (http://scholar.google.com), and 120 the China National Knowledge Infrastructure (http://cnki.net), using the keywords 121 'amino sugars', 'microbial necromass', 'microbial residue', 'fungal residue', and 122 123 'bacterial residue'. Data compilation was conducted following four stringent criteria to reduce bias from selected publications: (1) fungal and bacterial necromass (or 124 glucosamine [GluN] and muramic acid [MurA]) had to be reported simultaneously for 125 the same samples; (2) only data from agricultural and natural ecosystems (i.e., 126 grasslands and forests) were used; (3) for natural ecosystems, we excluded the data 127 from fertilized, polluted, treated, or anthropogenically disturbed sites; and (4) we only 128 included data from the top 0-20 cm of the soil profile reported in the publications, 129

| 130 | with other depths or unspecified soil depths excluded from our dataset. In total, the           |
|-----|-------------------------------------------------------------------------------------------------|
| 131 | final dataset consisted of 2094 observations from 486 sites worldwide (Figure 1)                |
| 132 | reported in 164 peer-reviewed papers. Among them, there were 1001 observations                  |
| 133 | from agricultural ecosystems and 1093 observations from natural ecosystems.                     |
| 134 | We calculated the FNC and BNC based on the contents of fungal GluN and                          |
| 135 | bacterial MurA using the following equations (Liang et al., 2019):                              |
| 136 | $FNC = \left(\frac{GluN}{179.17} - 2 \times \frac{MurA}{251.23}\right) \times 179.17 \times 9,$ |
| 137 | $BNC = MurA \times 45,$                                                                         |
| 138 | where 9 is the conversion factor from GluN to FNC, and, 179.17 and 251.23 are                   |
| 139 | the molecular weights of GluN and MurA, respectively; 45 is the conversion factor               |
| 140 | from MurA to BNC.                                                                               |
| 141 | Additional information including site geographic location (latitude and longitude),             |
| 142 | topographical condition (elevation), climatic factors (mean annual temperature [MAT])           |
| 143 | and mean annual precipitation [MAP]), soil physicochemical properties (pH, SOC,                 |
| 144 | total nitrogen (TN), clay content, and soil temperature), and biotic (microbial and             |
| 145 | plant) factors were recorded. Specifically, biotic factors included microbial biomass           |
| 146 | carbon (MBC), microbial biomass nitrogen (MBN), MBC/MBN, net primary                            |
| 147 | production (NPP), and belowground biomass C density (BGBC). The data of                         |
| 148 | topographical condition (elevation) was classified as geographical factor in this study.        |
| 149 | When MAT and MAP were unavailable in the original articles, we extracted them                   |
| 150 | from the global climate layers of WorldClim ( <u>http://www.worldclim.org/</u> ) with a grid    |

151 precision of  $30 \times 30$  arc sec according to geographic location. Missing elevation data

| 152 | were extracted using the <i>elevatr</i> package v.0.4.2 (Hollister, 2021) in the R    |
|-----|---------------------------------------------------------------------------------------|
| 153 | environment. We acquired the data on annual mean soil temperature from the study of   |
| 154 | Lembrechts et al. (2022), while other absent soil physicochemical data were extracted |
| 155 | from the Harmonized World Soil Database                                               |
| 156 | (https://www.fao.org/soils-portal/data-hub/soil-maps-and-databases/harmonized-world   |
| 157 | <u>-soil-database-v12/en/</u> ) and SoilGrids 2.0 (Poggio et al., 2021;               |
| 158 | https://www.soilgrids.org/) using ArcGIS 10.3. In addition, the data on NPP and       |
| 159 | BGBC were acquired from the studies of Zhao and Running (2010) and Spawn et al.       |
| 160 | (2020), respectively. Missing MBC and MBN data were acquired using a global           |
| 161 | database with a high resolution of $30 \times 30$ arc sec (Wang et al., 2022).        |

162

#### 163 2.2 Statistical analysis

All the statistical analyses were performed using R v4.1.3 (R Core Team, 2021). 164 Initially, the Shapiro-Wilk test was employed to assess the normality of our data, 165 followed by the application of Levene's test to evaluate the homogeneity of variances 166 across different groups. To detect the significant differences in the contributions of 167 FNC and BNC to SOC and the FNC/BNC ratio between agricultural and natural 168 ecosystems, as well as between forest and grassland ecosystems, the Wilcoxon rank 169 sum test was conducted. We used Spearman's rank correlation coefficient to explore 170 the connections between the 16 variables considered in this study, including 171 geographical and climatic factors, soil physicochemical properties, and biotic factors. 172 Since there was a strong positive correlation between MAT and soil temperature 173

(Figure S1), soil temperature was excluded from our subsequent analyses. Linear
regressions between different factors and the contributions of FNC and BNC to SOC
and the FNC/BNC ratio were performed. Dots and smoothing curves were drawn
using the *geom\_point* and *geom\_smooth* functions, respectively, in the *ggplot2*package v.3.4.0 (Wickham, 2016).

Variation partitioning analysis was conducted using the vegan package v.2.5.7 179 180 (Oksanen et al., 2020) to evaluate the effects of four types of factors on the contributions of FNC and BNC to SOC and the FNC/BNC ratio in agricultural and 181 182 natural ecosystems at global scale. We used a variance inflation factor threshold of 3.3 to eliminate those variables that were strongly correlated and avoid multicollinearity 183 (Figure S2; Kock, 2015; Fanin et al., 2020). Following factor selection, boosted 184 185 regression trees (BRTs) were used to partition independent influences of geographical (elevation) and climatic (MAT and MAP) factors, soil physicochemical properties (pH, 186 clay, C/N, and SOC), and biotic factors (NPP, BGBC, MBC, and MBC/MBN) on the 187 contributions of FNC and BNC to SOC and the FNC/BNC ratio with the gbm package 188 189 v.2.1.8.1 (Greenwell et al., 2022).

Utilizing the selected factors, we performed structural equation models (SEMs) to quantify the effects (direct, indirect and both) of four types of factors on the contributions of FNC and BNC to SOC and the FNC/BNC ratio using *lavaan* package v.0.6.19 (Rosseel, 2012). According to the previously reported potential causal relationships between explanatory and response variables (Wang et al., 2021a, 2021b; Li et al., 2024), we established the *priori* structural equation models for agro- and

natural ecosystems, respectively (Figure S3). The SEMs were fitted via maximum 196 likelihood estimation, with non-significant paths iteratively pruned through stepwise 197 exclusion, followed by model evaluation using modification indices and 198 goodness-of-fit criteria. The fit indices included degrees of freedom (df), chi-square 199 200  $(\chi^2, 0 \le \chi^2/df \le 2)$ , comparative fit index (CFI > 0.9), and root mean square error of approximation (RMSEA < 0.08), which were used to assess the adequacy of the SEM. 201 202 Map, box, bar, bubble, and lollipop charts were plotted with the ggplot2 package v.3.4.0 (Wickham, 2016). To enhance map visualization, the ggnewscale package 203 204 v.0.4.8 (Campitelli, 2022) was necessary alongside the ggplot2 package v.3.4.0 205 (Wickham, 2016). Similarly, the ggpubr package v.0.5.0 (Kassambara, 2022) was an additional necessity when creating lollipop charts. 206

207

#### 208 3 Results

- 3.1 Contributions of FNC and BNC to SOC and their ratio in agricultural and naturalecosystems
- There were no significant differences in the contributions of FNC and BNC to SOC and the FNC/BNC ratio between forest and grassland ecosystems (P > 0.05; Figure S4). Specifically, FNC contributed, on average, 29.11% to SOC in forests and 26.75% in grasslands, while BNC contributed 13.48% in forests and 14.34% in grasslands (Table 1). The average FNC/BNC ratios for forests and grasslands were 2.80 and 3.58, respectively.

217 In contrast, our analysis revealed statistically significant disparities in the

| 218 | contributions of FNC and BNC to SOC and the FNC/BNC ratio in agricultural and          |
|-----|----------------------------------------------------------------------------------------|
| 219 | natural ecosystems at the global scale ( $P 

(Figures 3b, d). Conversely, geographical factors, rather than soil physicochemical
factors, were the most important contributors to explain the FNC/BNC ratio in both
agricultural and natural ecosystems, accounting for 21% and 10% of the explained
variance in the FNC/BNC ratio, respectively (Figures 3e, f).

244 These findings were further substantiated by the results obtained from the BRTs. These results indicated that soil physicochemical factors accounted for a substantial 245 246 portion of the variance in the contributions of FNC and BNC to SOC in agricultural 247 and natural ecosystems (Figures 4a-d), and geographical factors played a similar role 248 in explaining the FNC/BNC ratio (Figures 4e, f). More precisely, soil physicochemical factors were identified as the primary contributors for the 249 contributions of FNC and BNC to SOC, with their contributions amounting to 51% 250 251 and 44% in agricultural ecosystems (Figures 4a, c), and 44% in natural ecosystems 252 (Figures 4b, d), respectively. This underscores soil physicochemical factors as the primary influencers on the contributions of FNC and BNC to SOC in both ecosystems. 253 Similarly, in the BRT models, geographical factors emerged as the primary 254 255 influencers of the FNC/BNC ratio in agricultural and natural ecosystems, accounting for 32% and 44% of the variance in each case, respectively (Figures 4e, f). In both 256 ecosystems, the BRT models used to quantify the relative influence of the four types 257 of factors on the contributions of FNC and BNC to SOC and the FNC/BNC ratio were 258 all significant (P 

| 262 | response of the contributions to individual soil physicochemical factors was not      |
|-----|---------------------------------------------------------------------------------------|
| 263 | entirely consistent between agricultural and natural ecosystems. Specifically, in the |
| 264 | BRT models, the C/N ratio was the third most influential factor, following clay and   |
| 265 | SOC, influencing the contribution of FNC to SOC in agricultural ecosystems (Figure    |
| 266 | 4a). However, the C/N ratio emerged as the most important factor influencing the      |
| 267 | contribution of FNC to SOC in natural ecosystems and the contribution of BNC to       |
| 268 | SOC in both ecosystems (Figures 4b-d). Linear regression models indicated that the    |
| 269 | contributions of FNC and BNC to SOC decreased with increasing C/N ratio in both       |
| 270 | ecosystems (Figures S5g, S6g). Elevation was the most significant geographical        |
| 271 | factors influencing the FNC/BNC ratio in both ecosystems (Figures 4e, f). Moreover,   |
| 272 | the FNC/BNC ratio in agricultural and natural ecosystems show significantly           |
| 273 | increased with an increase elevation (Figure S7a).                                    |

The SEMs revealed similar results, indicating that soil physicochemical factors 274 were the most influential factors for the contributions of FNC and BNC to SOC in 275 276 both agricultural and natural ecosystems, and the factor can affect the contributions of FNC and BNC to SOC in two ecosystems both directly and indirectly (Figures 5a-d, 277 6a-d). Notably, the direct and standardized total effects of soil physicochemical 278 279 factors on the contribution of FNC to SOC in natural ecosystem were both positive (Figures 6a, b). Geographical factors were the most influential factors for the 280 FNC/BNC ratio in agricultural and natural ecosystems, exerting both direct and 281 indirect effects on this ratio (Figures 5e, 6e), with the standardized total effect being 282 positive (Figures 5f, 6f). 283

284

### 285 4 Discussion

| 286 | MNC is an important component of SOC (Ma et al., 2018), and its variations can        |
|-----|---------------------------------------------------------------------------------------|
| 287 | influence the feedback effects on the C cycle and global climate change (Zhao et al., |
| 288 | 2023). Our study indicated that the contributions of FNC to SOC were approximately    |
| 289 | twice those of BNC in agricultural and natural ecosystems. Although the contributions |
| 290 | of FNC and BNC to SOC were significantly higher in agricultural ecosystems than in    |
| 291 | natural ecosystems, the FNC/BNC ratio was significantly higher in the latter. In      |
| 292 | addition, soil physicochemical properties and geographical factors were the most      |
| 293 | important predictors of the contributions of MNC (FNC and BNC) to SOC and the         |
| 294 | FNC/BNC ratio in the two ecosystems, respectively. These findings enhance our         |
| 295 | understanding of microbially mediated biogeochemical cycling processes under          |
| 296 | current and future climate scenarios.                                                 |

297

4.1 Variation in the contributions of FNC and BNC to SOC and the FNC/BNC ratio inagricultural and natural ecosystems

With growing appreciation for the critical role of microbial necromass constituents in forming SOC, it is critical to understand the different drivers of this process across the globe. In this study, we found that the contribution of FNC to SOC significantly exceeded that of BNC, with the former contributing approximately twice as much as BNC to SOC in both agricultural and natural ecosystems (including forests and grasslands; Table 1). These findings were consistent with previous studies (Lian