# Peer review of "Soil microbial necromass shapes global carbon stocks in agricultural and natural"

_Earth System Science Data, 2025_

## Author Comment (AC1)

Dear Editor,

We would like to thank you, and the reviewers for the contributions to this manuscript.

The constructive feedback has been extremely helpful. We have accepted the vast

majority of the changes suggested and made the appropriate changes to the study. We

believe that the manuscript is considerably clearer and more impactful as a result.

Attached please find our point-by-point responses to the reviewer's comments.

We thank you for your consideration and hope you will find this version suitable for

publication in Earth System Science Data.

Best regards,

Zhiqiang Wang, and on behalf of all co-authors

Sichuan Zoige Alpine Wetland Ecosystem National Observation and Research Station,

Southwest Minzu University

Chengdu, 610041, PR China

E-mail: wangzq@swun.edu.cn

**Response to reviewer's comments**

**Responses to the Reviewer's comments**

The study worked out the fungal and bacterial necromass contribution to SOC and interpreted the variability with climate, geographical and soil conditions. The data are useful and provide key reference for global ecosystem study of SOC storage. However, as the authors often mentions their work was very similar, or consistent, with others work already published. The shortage may be not robust information about their soil samples, particularly agricultural soils not classified and sampling condition not clearly defined. It could be improved if the authors could add samples from managed grasslands, wetlands and divided agricultural soil into dry land and irrigated as well as submerged paddy. The discussion need rewrite, reorganized and better presented with statistical analysis.

**Response:** Thank you for your positive and constructive feedback.

We also apologize for any confusion we may have caused to this reviewer. Our results, not only agree with previous work (e.g., fungal necromass-C patterns), they also provide new unreported insights. For instance, we found that the contributions of both fungal (FNC) and bacterial (BNC) necromass to soil organic carbon (SOC) were significantly higher in agricultural ecosystems compared with natural ecosystems. Moreover, we further identified soil properties (e.g., soil C/N ratio and clay content) as the primary drivers of FNC and BNC contributions to SOC. This knowledge is critical for supporting carbon sequestration in terrestrial ecosystem.

Here, in response to the reviewer's comment, we used Google Earth Engine and the LGRIP30 V1 dataset to classify agricultural ecosystems into dryland and irrigated cropland. We further applied the JRC surface-water seasonality layer to identify submerged paddy fields within the irrigated class. This process categorized the 145 agricultural ecosystem samples into 32 dry land, 72 irrigated, and 41 submerged paddy sites (Lines 155–157 of the revised manuscript). Subsequent analysis revealed that the contributions of FNC and BNC to SOC were comparable between dryland and irrigated systems but significantly lower in submerged paddy fields, while the FNC/BNC ratio was similar across all three subtypes (Lines 202–204, 256–261, 271–273; Figure S4). We attribute this to the anaerobic conditions in paddy soils suppressing aerobic decomposition and altering microbial processes, thereby reducing overall necromass accumulation without shifting the ratio between FNC and BNC (Qiu et al., 2017; Chen et al., 2021). However, the resulting sample sizes were insufficient for more advanced statistical modeling, so only non-parametric tests were applied to these subgroups.

Figure S4. Comparison of the contributions of MNC to SOC, and their ratios among dry land, irrigated cropland and submerged paddy in this study. Comparison of the contributions of FNC (a) and BNC (b) to SOC, and FNC/BNC ratio (c) among dry land, irrigated cropland and submerged paddy. The same capital letter in the same panel indicates that there is no significant difference among the groups (P > 0.05), while different capital letters indicate that there is a significant difference among the groups (P < 0.05).

Moreover, we have reorganized and substantially revised the Discussion section. For further details regarding the results and their interpretation, please refer to lines 316–476 of the revised manuscript. We hope that all concerns have been adequately addressed and the overall quality of the manuscript has been significantly improved. Below, we provide point-by-point responses to your comments.

**Specific comments as follows:**

Title: Suggest to change as "Microbial necromass contribution to topsoil organic carbon storage of natural and agricultural ecosystems".

Response: Done.

**INTRO**

Line 47-51: Soil organic matter, the material containing organic carbon as the core element but preserved in soil matrix, is truly formed based on plant biomass (material), even the organic pollutants in soil based on fossil plant material (coal and petro-oil).

It is not contradictory that SOM is composed of plant derived and microbes-derived organics, the letter is formed of metabolic residues upon microbial processing of plant material and preserved through interaction with mostly soil mineral matrix. That is to say, microbial necromass is indirectly derived from plant material, or microbial products via processing plant material. Both plant derived or microbial derived organic carbon could be stable in soil conditions preservation could be allowed. "Stable" here could be changed into "with long turnover time".

**Response:** We thank the reviewer for this important clarification. We fully agree that plant biomass is the ultimate source of carbon entering soils and that microbially derived organic matter—including microbial necromass—is indirectly plant-derived, resulting from microbial processing of plant inputs. Its preservation is predominantly governed by interactions with the soil mineral matrix. To avoid any unintended plant-microbe dichotomy, we have reframed the description of these pools as successive stages along a continuum: from plant input, to microbial transformation, to mineral-associated preservation.

As suggested, we have revised the relevant statement to now read:

"In brief, plant inputs provide the primary carbon source to soils, and microbial processing transforms these inputs into microbial necromass that can persist over long turnover times (Angst et al., 2021; Cotrufo et al., 2013)."

For further detail, please see Lines 45–47 of the revised manuscript.

Line 55-58: Microbial biomass carbon generally possessed generally about 2% to SOC, but microbial necromass could predominate (as high as 80% to SOC) in soils low in SOC. For the general estimation (2%) of microbial biomass C percentage to SOC (often termed microbial quotient), maybe cite a recent study (Topsoil microbial biomass carbon pool and the microbial quotient under distinct land use types across China: A data synthesis, Soil Science and Environment, 2:5. Doi:10.48130/SSE-2023-0005).

**Response:** Thank you for this constructive suggestion. In the Introduction, we have now clearly defined the microbial quotient (i.e., microbial biomass carbon as a percentage of soil organic carbon, MBC/SOC) and supported this by citing both a classical reference and the recommended recent synthesis. We have also qualified our statement regarding the dominance of microbial necromass with up-to-date evidence. Specifically, the revised text now reads:

"Although the living soil microbial biomass typically constitutes only about 2% of SOC (a ratio referred to as the microbial quotient; Anderson & Domsch, 1989; Liu et al., 2023), microbial necromass has been shown to contribute more than half and up to approximately 80% of SOC, depending on soil type and analytical methods (Liang & Balser, 2011; Kallenbach et al., 2016; Liang et al., 2019)."

For further details, please see Lines 48–52 of the revised manuscript.

Line 60-63: many studies...., add description of the study conditions such as land use,

management, climate change, regional, etc.

**Response:** Following the constructive comment, we have reorganized and thoroughly revised this section in the manuscript. The updated text now reads:

To gain a comprehensive understanding of MNC in the global C cycle, recent research has highlighted the distinct roles of fungal and bacterial necromass, revealing their contrasting responses to environmental and anthropogenic drivers. For instance, studies have shown that the accumulation and contribution of MNC are sensitive to factors such as aridity, primary productivity, agricultural management practices like tillage and fertilization, as well as key soil properties including pH and clay content (Zhang et al., 2021; Zhou et al., 2023; Xu et al., 2024). Despite these advances, it remains unclear whether these organism-specific mechanisms translate into systematic differences in necromass contributions between ecosystems under varying degrees of human interference, such as agricultural versus natural systems.

For further details, please refer to Lines 58–68 of the revised manuscript.

Line 64-65: With the distinct roles of fungi and bacteria in decomposing organic matter and stabilizing organic carbon in soil, the relative contribution to SOC of fungal and bacterial necromass C could be used to track the dynamics of SOC storage (Malik et al., 2016).

Response: Done.

Line 66-68: Vague sentence, suggest to delete.

Response: Done.

Line 71-74: As bacterial amino-sugars is degradable rather than fungal chitin or  $\beta$ -glucans, fungal necromass existed in soil generally with longer turnover time than bacterial necromass.

**Response:** Done.

Line 74-78: The sentence may not be correct here. By definition, microbial necromass is not microbial biomass.

**Response:** The statement has been revised in the manuscript as follows:

Wang et al. (2021a) reported that the contribution of fungal necromass carbon (FNC) to SOC exceeded 65%, considerably higher than that of bacterial necromass carbon (BNC, 32–36%). This pattern is likely attributed to the slower decomposition rate and stronger mineral-associative capacity of fungal necromass. Furthermore, greater fungal biomass and higher turnover rates may enhance the input flux of fungal necromass (Klink et al., 2022).

For further details, please refer to Lines 77–82 of the revised manuscript.

Line 78-82: Pls delete "Previous studies have also indicated that".

Response: Done.

Line 82-86: May be changed into "However, few studies on fungal and bacterial necromass carbon and their contribution to SOC has been reported for ecosystems under human interference (Chen et al., 2020)". The citation of (Chen et al., 2020) may not be appropriate here.

Response: Done.

Line 88-90: Pls consider to change. The agricultural ecosystems are typical of plant litter derived of single crops under human management (Bohan et al., 2013).

**Response:** Thanks for this comment. We have rewritten in the revised manuscript (Lines 88–90).

Line 90-92: In contrast, natural ecosystems display greater diversity in plant litter and root deposits (Wu et al., 2019).

**Response:** Thank you for the suggestion. We have rewritten in the revised manuscript (Lines 91–92).

*Line* 95-98: in comparison to bacterial?

**Response:** Thank you very much for this thoughtful comment. The Reviewer raised an excellent point, and we agree that a comparative perspective with bacterial decomposers provides valuable context and significantly strengthens our argument. The relevant statement has been revised as follows:

While bacteria are undoubtedly vital decomposers, fungi play a distinct and often dominant role in the initial breakdown of complex plant polymers such as cellulose and lignin. This functional prominence stems from their potent enzymatic machinery and hyphal growth form, which enable physical penetration and decay of solid organic matter (de Boer et al., 2005). As key decomposers, fungi are thus critical in processing cellulose and other complex organic compounds (Hättenschwiler et al., 2005). Accordingly, as demonstrated by Choi et al. (2018), soil cellulose-degrading genes are frequently linked to fungal activity and abundance.

For further details, please refer to Lines 95–103 of the revised manuscript.

Line 98-102: These statements may be of questions, or not sufficient. For example, why diverse plant input lead to rich soil cellulose content, why not diverse SOM composition?

**Response:** Thank you for this constructive comment. We fully agree that diverse plant inputs do not necessarily imply higher cellulose concentration per se, but rather a broader spectrum and higher heterogeneity of plant-derived polymers—such as cellulose, hemicelluloses, and lignin—along with other compounds. This increased chemical diversity can broaden decomposer niches and often favors fungal taxa in litter horizons. Empirical syntheses indicate that plant/litter diversity generally enhances fungal diversity and shifts decomposer community composition, whereas agricultural monocultures tend to reduce fungal diversity—though these effects can be moderated

by management practices such as conservation tillage and organic amendments. Furthermore, while cellulose additions or cellulose-rich inputs can enrich saprotrophic fungi in arable soils, bacteria may also contribute substantially in mineral soils or under specific environmental conditions.

We have revised the relevant statement to reflect this more nuanced, context-dependent perspective, emphasizing diverse soil organic matter composition rather than cellulose abundance alone:

"Rather than implying higher cellulose concentration per se, diverse plant inputs increase the chemical heterogeneity of plant-derived polymers (e.g., cellulose, hemicelluloses, and lignin), which broadens decomposer niches and often favors fungal communities in litter horizons (Hättenschwiler et al., 2005; Štursová et al., 2012). In contrast, agricultural monocultures tend to reduce fungal diversity unless mitigated by management practices (Chen et al., 2020). Reflecting this context dependence, cellulose-rich inputs can enrich saprotrophic fungi in arable soils (Clocchiatti et al., 2021), whereas bacteria may contribute substantially in mineral soils or under specific microhabitat and land management conditions (Štursová et al., 2012; Choi et al., 2018)." For further details, please refer to Lines 103–112 of the revised manuscript.

Line 102-106: The issue is important, but the rational is not strong as your overview of studies. Could you focus on why FNC and BNC across global ecosystems or why the ratio is important across the global ecosystems?

**Response:** Thank you for this constructive comment. We agree that the rationale should clearly articulate the global-scale significance of fungal necromass carbon (FNC), bacterial necromass carbon (BNC), and their ratio (FNC/BNC). Accordingly, we have revised the Introduction to emphasize three key points:

- (1) Fungal and bacterial necromass differ fundamentally in cell-wall chemistry (e.g., chitin/β-glucans vs. peptidoglycan) and pathways of organo-mineral association, resulting in contrasting turnover times and stabilization mechanisms. As such, the FNC/BNC ratio serves as an integrative indicator of decomposition pathways and soil organic carbon (SOC) formation (Kleber et al., 2021; Angst et al., 2021).
- (2) The relative accumulation of FNC and BNC—and thus their ratio—varies systematically with climatic and edaphic factors (e.g., aridity, mean annual temperature, pH, and clay content) and land management practices (e.g., tillage and fertilization), as demonstrated across grassland, cropland, and forest ecosystems (Zhang et al., 2021; Zhou et al., 2023; Xu et al., 2024).
- (3) Since management practices can differentially influence fungal and bacterial necromass (e.g., fertilization often elevates bacterial residues), the FNC/BNC ratio provides a concise yet powerful metric for predicting management-specific SOC dynamics and improving microbial-explicit SOC models (Zhou et al., 2023).

The corresponding statement in the revised manuscript (Lines 112–119) now reads: "Due to distinct chemical properties and organo-mineral stabilization pathways, fungal and bacterial necromass exhibit differing turnover times, making the FNC/BNC ratio a mechanistic tracer of SOC formation (Angst et al., 2021; Kleber et al., 2021). Therefore, elucidating the global distribution and drivers of FNC, BNC, and their ratio across agricultural and natural ecosystems is essential for predicting management-induced shifts in SOC under varying climatic and soil conditions (Zhang et al., 2021; Zhou et al., 2023; Xu et al., 2024)."

*Line 112-113: delete the sentence as is already ascertained above.*

Response: Done.

MM

Line 117-118: to be deleted.

**Response:** Done

Line 119: by December 31, 2021?

**Response:** We sincerely apologize for the confusion caused by our imprecise phrasing. In fact, we systematically screened and curated peer-reviewed publications up to 31 December 2022 and assembled a comprehensive global dataset. The considerable effort required for data integration and subsequent analyses unfortunately delayed our submission. We have now corrected the text accordingly in the revised manuscript (Line 130).

Line 121-123: why not fungal derived glomalin-related proteins? And, why "fungal necromass, bacterial necromass included in your search engine?

Response: Thank you for this thoughtful comment. Our synthesis specifically aimed to quantify fungal necromass carbon (FNC), bacterial necromass carbon (BNC), and their ratio (FNC/BNC) using comparable and stoichiometrically grounded proxies. Accordingly, our search strategy targeted studies that reported amino sugar biomarkers—glucosamine (GluN) and muramic acid (MurA)—from which FNC and BNC can be derived using established conversion factors (as provided in Equation 1 and Equation 2 in the Materials and Methods). To capture variations in terminology while maintaining specificity, we also included keyword pairs such as "fungal residue/necromass" and "bacterial residue/necromass," alongside broader terms like "amino sugars" and "microbial residue/necromass." This approach ensured the inclusion of studies that concurrently report fungal and bacterial residues in the same samples, as required by our inclusion criteria.

Regarding glomalin-related soil protein (GRSP), we intentionally excluded it from our search terms for the following methodological reasons:

1. Operational Definition and Mixed Provenance: GRSP is operationally defined through citrate-autoclave extraction and often co-extracts non-arbuscular

- mycorrhizal fungal proteins, lipids, and humic substances (Rillig, 2004). It is therefore not a taxon-specific necromass proxy, and its biochemical identity remains contentious.
- 2. Lack of Standardized Conversion and Partitioning: GRSP measurements cannot be disaggregated into fungal versus bacterial components, and there are no universally accepted stoichiometric conversion factors to estimate necromass carbon. This precludes the calculation of FNC, BNC, and FNC/BNC, which are central to our analysis (Irving et al., 2021).
- 3. Incompatibility with Inclusion Criteria: Our study requires simultaneous reporting of GluN and MurA (or directly derived FNC and BNC) from the same samples. Studies focusing solely on GRSP generally do not meet this criterion and would introduce methodological heterogeneity incompatible with amino sugar-based estimates.

We sincerely hope this clarification adequately addresses your concern.

Line 123-130: studies search and screening procedure are not well described. I suppose you first collected all the studies indexed of the keywords, then you made a rough compilation. Secondly, filter the compiled studies with "topsoil"; followed by paired data of fungal and bacterial necromass or so, further divided your filtered studies into ecosystem categories, finally you excluded those potentially disturbed ecosystems from the natural category. Pls do organize clearly your work flow and display in a flow chart or in an order of steps.

**Response:** Thank you very much for this helpful suggestion. We have reorganized the "2.1 Data Collection" section of the Materials and Methods to explicitly follow the recommended sequence and present a clear, stepwise workflow:

- (1) We collected peer-reviewed papers published from 1996 to 31 December 2022 from Web of Science (http://apps.webofknowledge.com), Google Scholar (http://scholar.google.com), and the China National Knowledge Infrastructure (http://cnki.net), using the keywords: 'amino sugars', 'microbial necromass', 'microbial residue', 'fungal residue', and 'bacterial residue'. Records from different databases were merged and deduplicated to form an initial compilation.
- (2) We then filtered the compiled studies to include only those focusing on topsoil, defined as the 0–20 cm layer. Studies reporting deeper or unspecified sampling depths (e.g., 0–30 cm) were excluded to ensure spatial comparability.
- (3) Full texts were assessed to confirm the presence of paired fungal and bacterial residue data from the same sample—specifically, glucosamine (GluN) and muramic acid (MurA), or directly reported FNC and BNC values—to enable consistent cross-study calculation of the FNC/BNC ratio. Studies lacking either biomarker were excluded from ratio analyses, though those directly reporting the FNC/BNC ratio were retained.
- (4) Eligible observations were classified into agricultural ecosystems (including dry land, irrigated cropland, and submerged paddy) or natural ecosystems (forest and grassland) based on study metadata.

(5) For natural ecosystems, data from fertilized, polluted, experimentally treated, or otherwise anthropogenically disturbed sites were excluded.

For further details, please refer to Lines 129–147 of the revised manuscript.

A question, here you claim that only data of topsoil (0-20cm) were collected. As I experienced, different depth intervals of topsoil were used for agroecosystems and natural ecosystems, mostly 0-30cm for dry croplands while often undiscerned for natural ecosystems. You may mention these different usages though not critical for your relative contribution and ratio estimation. But this may affect estimation of mass abundance of microbial necromass in soil.

**Response:** Thank you very much for raising this important point. In our synthesis, we harmonized the topsoil depth to 0–20 cm to maximize comparability across ecosystems and studies. Records falling outside this range (e.g., 0–30 cm or unspecified depths) were excluded to prevent depth-related biases.

As the reviewer rightly noted, we further clarify that our primary metrics—namely, the contributions of fungal necromass carbon (FNC) and bacterial necromass carbon (BNC) to soil organic carbon (SOC), and the FNC/BNC ratio—are dimensionless or concentration-based, and are therefore less sensitive to moderate variations within the topsoil layer. In contrast, mass-based abundance or stock estimates (e.g., area-based stocks) can be more strongly influenced by sampling depth (Von Haden et al., 2020; Wendt & Hauser, 2013).

For further details, please refer to Lines 135–138 of the revised manuscript.

Another key question: did you screen the data for collecting and measurement standard protocol? Is the similar season or growing period among all the samples? All the samples were treated with a consistent procedure (for example sample preparation, shipping and storage, and analysis condition).

**Response:** Thank you for your comment. Our data compilation specifically targeted amino sugar biomarker—based estimates of microbial necromass carbon (MNC), in which muramic acid (MurA) serves as the biomarker for bacterial necromass and glucosamine (GluN)—after correcting for its bacterial contribution—as the biomarker for fungal necromass. Following the measurement of amino sugar concentrations, fungal necromass carbon (FNC) and bacterial necromass carbon (BNC) were calculated using the established conversion formulas provided in Equation 1 and Equation 2 (Lines 161 and 167) of the revised manuscript. Accordingly, our search strategy explicitly included the keyword "amino sugars" (Line 133).

Regarding seasonal or growing period variations, as well as sample handling protocols (such as preparation, shipping, storage, and analytical conditions), we did not impose uniform requirements across the included studies. Given that this synthesis spans diverse continents and ecosystems, enforcing a single standard for these aspects would

have significantly limited the number of eligible records, thereby compromising the statistical power required for our cross-ecosystem analyses—including variance partitioning, boosted regression trees, and structural equation modeling. We sincerely hope this response adequately addresses your concerns.

Line 130-133: What about the time span of the published studies? How did you check if repeated data in publications of a same study but in different journals and/or years?

**Response:** Thanks for your comment. Our dataset encompasses studies published between 1996 and 2022. We have now explicitly included this time span in the revised manuscript (Line 130).

To mitigate the risk of duplicate reporting across different publications or years, we implemented a systematic duplication-control protocol during full-text screening and data extraction. Specifically, we constructed a unique study-site-year-depth identifier based on the first author, DOI or journal name, site name or coordinates (with a tolerance of  $\pm 0.01^{\circ}$ ), sampling year, sampling depth (0–20 cm), and land-use class. Records with overlapping identifiers and nearly identical statistical values (e.g., means, standard errors, sample sizes) or shared project identifiers were flagged for further evaluation.

In cases of overlapping publications, we retained the primary or most comprehensive source—specifically, the article reporting paired FNC and BNC values (or glucosamine and muramic acid concentrations) along with variance measures and complete metadata. Secondary or duplicate reports were excluded unless they introduced non-overlapping experimental treatments or distinct sampling years. When the sampling year was not explicitly stated, we extracted it from the main text or supplementary materials. If this information was unavailable, we used the publication year as a provisional proxy and marked these entries for sensitivity analysis.

*Line 134-137: Pls describe the calculation of FNC and BNC respectively, and number the equations.*

**Response:** Thanks for the helpful suggestion. We have revised in the manuscript (Lines 158–169).

*Line 138-140: Pls clarify the unit of the calculated contents.*

**Response:** Thanks for the helpful suggestion. We have clarified this in the revised manuscript (Line 162–166, 168–169).

Line 141-145: How did you obtain the information of soil temperature. As I know well, soil temperature is not normally recorded while in field sampling. However, soil moisture content data could be available in most sampling procedure or lab measurement before further analysis for specific purposes. In addition, what kind of information for microbial or plant factors.

**Response:** Thanks for your comment. Soil temperature is indeed not routinely measured during field sampling. In our synthesis, the annual mean soil temperature for each site was extracted from the global soil temperature maps provided by Lembrechts et al. (2022) based on site coordinates. This procedure is now explicitly described in the Materials and Methods (Section 2.1 Data Collection; Lines 181–182). To avoid redundancy and multicollinearity, we assessed correlations among all predictors and, due to a strong positive correlation between soil temperature and mean annual temperature (MAT; Figure S1), soil temperature was excluded from subsequent analyses. This is clarified in the Materials and Methods (Section 2.2 Statistical Analysis; Lines 207–209).

Regarding soil moisture, we agree that it serves as an ecologically important driver. However, instantaneous soil moisture at the time of sampling was inconsistently reported across studies, which precluded its use as a harmonized predictor in our global models without substantially compromising sample size and representativeness. As an alternative, we used mean annual precipitation (MAP) and soil texture (clay content) as proxies for site-level moisture regime and water-holding characteristics, as these variables are consistently available at a global scale and capture a substantial portion of moisture-related variability.

Concerning "microbial or plant factors," we have now explicitly listed the variables retained in our analysis: microbial biomass carbon and nitrogen (MBC and MBN) and their ratio (MBC/MBN), as well as net primary production (NPP) and belowground biomass carbon density (BGBC). Data sources and retrieval methods for these variables are detailed in the Materials and Methods (Section 2.1 Data Collection; Lines 190–193).

Line 151: what is the spatial distance of 30  $\times$  30 arc sec? Is such grid resolution comparable to the site specific climate data?

**Response:** Thank you for this constructive comment. In WorldClim, a spatial resolution of  $30 \times 30$  arc seconds corresponds to approximately  $0.008333^{\circ}$ . This translates to roughly 0.93 km in the north–south direction globally, and about  $0.93 \times \cos(\varphi)$  km in the east–west direction (approximately 0.66 km at  $45^{\circ}$  latitude).

In our synthesis, climatic variables were extracted at the reported site coordinates from the  $30 \times 30$  arc second (approximately 1 km) grids provided by WorldClim. This resolution is standard in global syntheses and is well suited to represent the macroclimatic context at plot to site scales. Although gridded climate products cannot fully capture microclimatic variability at the exact sampling location, we addressed potential mismatches in two ways:

(1) Our primary response variables—namely the FNC/BNC ratio, the contribution of FNC to SOC, and the contribution of BNC to SOC—are dimensionless or concentration-based. These metrics are comparatively less sensitive to modest

- variations in within-topsoil conditions and microclimate than area-based stock estimates.
- (2) We explicitly incorporated elevation, soil texture (clay content), and other covariates that co-vary with local temperature and moisture regimes, thereby helping to account for sub-grid environmental heterogeneity.

Line 153-158: Use of data of soil temperature and soil properties digested from the GEO-based data base is questionable for the studied soil in your database.

**Response:** Thanks for this constructive comment. We fully agree that gridded (GEO-based) data must be used judiciously in soil-related studies. As a synthesis effort, however, not all original studies reported every variable of interest required for our analysis. In such cases, field-reported values were always prioritized, and gridded data were used solely to supplement missing covariates at the corresponding site coordinates. Specifically, when mean annual temperature (MAT) or mean annual precipitation (MAP) were unavailable, they were extracted from WorldClim at a  $30 \times 30$  arc second resolution ( $\sim$ 1 km). Annual mean soil temperature was obtained from Lembrechts et al. (2022), and missing soil physicochemical properties were retrieved from the Harmonized World Soil Database (HWSD) and SoilGrids 2.0, with values specifically drawn from topsoil layers matching the 0–20 cm depth (as detailed in Section 2.1 Data Collection; Lines 177–187).

To mitigate potential mismatches between macro- or meso-scale gridded data and plot-level conditions, soil temperature was retained for descriptive purposes but excluded from multivariate modeling due to its strong collinearity with MAT (Section 2.2 Statistical Analysis; Lines 207–209). Furthermore, after acquiring values from gridded sources, we applied a field-anchored bias correction—using site- or region-specific delta adjustments calibrated against available field measurements—to minimize errors introduced by the use of gridded data (Section 2.1 Data Collection; Lines 187–190).

Line 161-162: data of microbial biomass carbon and nitrogen is not eligible from the geo-database. These varies very much from site to site, or from time to time.

**Response:** Thank you for this insightful comment. We fully acknowledge that microbial biomass carbon (MBC) and nitrogen (MBN) exhibit considerable spatiotemporal variability, and we agree that in situ measurements represent the ideal source for such data.

In our global-scale analysis, the primary objective was to compile a consistent and comprehensive dataset to evaluate the drivers of microbial necromass across all available sites. For locations where MBC and MBN were not reported in the original publications, we supplemented these data using the high-resolution  $(30 \times 30 \text{ arc second})$  global gridded datasets from Wang et al. (2022). This database was selected for three main reasons:

- (1) It represents the most sophisticated and high-resolution global source for MBC and MBN, derived from machine learning models trained on an extensive compilation of over 25,000 field measurements;
- (2) The use of this standardized dataset enabled us to maintain a globally consistent set of covariates across all sites, which is essential for robust cross-ecosystem comparative analysis;
- (3) Since its publication in *Catena*, this database has undergone peer review and has been widely adopted and validated in multiple subsequent global-scale ecological studies (e.g., Han et al., 2024; Shi et al., 2024; Yao et al., 2024), underscoring its reliability and acceptance within the scientific community for macroecological applications.

We recognize that this approach introduces a degree of uncertainty. However, we emphasize that MBC and MBN were not the primary response variables in our study, but rather were included as covariates in multivariate models—including variation partitioning, boosted regression trees, and structural equation modeling—to account for potential biotic influences (as described in Section 2.2 Statistical Analysis; Lines 214–236). To address collinearity, we implemented a rigorous variable selection procedure using a variance inflation factor (VIF) threshold of 3, ensuring that only variables with independent explanatory power were retained in the final models (Section 2.2 Statistical Analysis; Lines 217–219).

**Results**

Line 224-226: How did you get these values? Calculation using the numbers you provided in the preceding sentences does not yield the same values (2.23 for agricultural but 2.09 for natural). If the calculation correct, there is significant but slight difference in FNC/BNC ratio between agricultural and natural ecosystems.

**Response:** Thank you for your professional and meticulous comment. As previously mentioned, this study is an integrative synthesis based on 2,094 observations drawn from 164 peer-reviewed articles. Since the original publications come from diverse journals and do not uniformly report a complete set of variables for each sample—and because we did not impute missing values for fungal necromass carbon (FNC) or bacterial necromass carbon (BNC)—some records contain only the FNC/BNC ratio, only FNC, or only BNC. As a result, back-calculations based on the numbers provided in the preceding sentences may not align with the values reported in Lines 224–226 of the manuscript.

The values cited in the manuscript (3.22 vs. 2.61) were computed directly from the underlying site-level data—specifically, from the available FNC/BNC ratios—rather than derived from the ratio of group means. To ensure full transparency, the complete raw dataset used for these calculations has been deposited in Figshare (DOI: 10.6084/m9.figshare.28827386).

We hope this clarification adequately addresses the reviewer's concern.

The samples of agricultural ecosystem not clearly defined. Dry croplands, irrigated croplands, rain-fed dry lands and waterlogged paddies? Also, the cultivation history is important, at least need to clarify those shortly shifted from natural ecosystem, for example from grassland.

**Response:** Thank you for this constructive comment. In accordance with the reviewer's suggestion, we used Google Earth Engine with the LGRIP30 V1 dataset to classify agricultural ecosystems into dry land and irrigated cropland. We further overlaid the JRC surface water seasonality layer to identify submerged paddy fields within the irrigated class (defined as LGRIP30 irrigated value = 2 and JRC seasonality ≥ 1). This process resulted in the classification of 145 agricultural ecosystem samples into 32 dry land sites, 72 irrigated cropland sites, and 41 submerged paddy sites (as detailed in Lines 152–157 of the revised manuscript). We then performed Kruskal–Wallis tests followed by Dunn's post hoc comparisons (Lines 202–204).

The results indicated that the contributions of fungal necromass carbon (FNC) and bacterial necromass carbon (BNC) to soil organic carbon (SOC) did not differ significantly between dry land and irrigated cropland (P > 0.05), though both differed markedly from submerged paddy systems (P < 0.05; Figure S4a, b; Lines 256–258). In contrast, the FNC/BNC ratio showed no significant differences among dry land, irrigated cropland, and submerged paddy (P > 0.05; Figure S4c; Lines 271–272).

This pattern may be attributed to similar aeration conditions in dryland and irrigated systems—both being predominantly oxygenated—which support comparable decomposition, transformation, and mineral-association pathways, ultimately leading to similar net contributions of fungal and bacterial residues to SOC (Ghezzehei et al., 2019). In submerged paddy soils, however, persistent or periodic flooding creates anoxic conditions that suppress aerobic decomposition and shift microbial metabolic pathways (e.g., toward denitrification and methanogenesis; Qiu et al., 2017). These changes likely reduce fungal activity or dominance and alter the relative accumulation and stabilization of fungal versus bacterial necromass, resulting in significantly lower contributions of both FNC and BNC to SOC compared to non-flooded systems.

Notably, although flooding can suppress fungi, it may also enhance the overall retention of both fungal and bacterial necromass through reduced decomposition rates and enhanced mineral protection, thereby preserving the FNC/BNC ratio even as absolute contributions decline (Chen et al., 2021; Gao et al., 2024; Lines 395–410).

After subdividing agricultural ecosystems into these three categories, the resulting sample sizes (dry land: 263; irrigated cropland: 634; submerged paddy: 104) were unfortunately insufficient to support more complex downstream analyses—such as variance partitioning, boosted regression trees, and structural equation modeling—with adequate statistical power. Therefore, aside from the non-parametric comparisons,

these subdivided categories were not included in subsequent multivariate analyses.

We fully agree that cultivation history is an important factor influencing agricultural soil properties. However, among the 164 peer-reviewed papers included in our synthesis, the majority did not report site-level cultivation history, or reported it in formats that were inconsistent and non-comparable across studies. Given this incomplete and heterogeneous reporting, we regret that we are unable to construct a consistent, synthesis-wide variable for cultivation history without introducing substantial bias or uncertainty. We sincerely hope this clarification is acceptable.

Figure S4. Comparison of the contributions of MNC to SOC, and their ratios among dry land, irrigated cropland and submerged paddy in this study. Comparison of the contributions of FNC (a) and BNC (b) to SOC, and FNC/BNC ratio (c) among dry land, irrigated cropland and submerged paddy. The same capital letter in the same panel indicates that there is no significant difference among the groups (P > 0.05), while different capital letters indicate that there is a significant difference among the groups (P < 0.05).

Line 228 the subheading of "Effects of the driving factors on..." may not be proper for this is a synthesis of data in arbitrary studies without certain treatments. Could be change into "Driving factors of the change in fungal and bacterial necromass contribution to SOC and their ratio". But this context should be presented in Discussion part, not the direct results presented here.

**Response:** Thanks. We have revised the subheading to "Driving factors of the change in fungal and bacterial necromass contribution to SOC and their ratio" and relocated the relevant content to 4 Discussion (4.2 Driving factors of the change in fungal and bacterial necromass contribution to SOC and their ratio). For more details, please refer to Lines 412–413 of the revised manuscript.

I suggest you could split your result into two subheadings: 3.1 Fungal and bacterial necromass contribution to SOC; 3.2 Ratio of Fungal and bacterial necromass. In 3.1, you may provide more detailed information of the variation of fungal and bacterial necromass content and the contribution to SOC, among samples, ecosystem types and or other dimension (for example, regionally). In 3.2, provide the ratio variance among the systems, but also digest the relations to SOC level. Possibly, you could align your correlation to these variance to digest the driving factors, respectively.

**Response:** Thank you for this constructive suggestion. As recommended, we have reorganized the Results section into two distinct subsections:

- 3.1 Fungal and Bacterial Necromass Contributions to SOC in Agricultural and Natural Ecosystems
- 3.2 Ratios of Fungal to Bacterial Necromass in Agricultural and Natural Ecosystems

In Section 3.1, we now provide detailed comparisons across agricultural subtypes (dryland, irrigated, and paddy) and natural ecosystems (forests and grasslands), emphasizing differences among these ecosystem types. It should be noted that, due to the absence of regional classification (e.g., by continent or climate zone) in our dataset, geographical variations in fungal and bacterial necromass contributions to SOC were not explicitly addressed in this study.

In Section 3.2, we present the FNC/BNC ratios across agricultural and natural ecosystems, including their respective subtypes. In accordance with the reviewer's suggestion and to maintain an objective presentation of results, the analysis and interpretation of the relationship between the FNC/BNC ratio and SOC level have been moved to the Discussion (Lines 374–394).

Additionally, to improve clarity, we have introduced a new subsection:

3.3 Associations of Abiotic and Biotic Factors with Microbial Necromass Parameters This section examines the relationships between both abiotic and biotic factors and microbial necromass parameters—including the contributions of fungal and bacterial necromass carbon to SOC, as well as the FNC/BNC ratio.

For further details, please refer to Lines 278–314 of the revised manuscript.

**Discussion**

This part not well organized, often repeating the statement of results.

**Response:** We thank the reviewer for this helpful comment. We have substantially

restructured and rewritten this part to focus on interpretation rather than restating results. For more details, please refer to Lines 316–476 of the revised manuscript.

*Line* 286-296: *Not a single independent paragraph.*

**Response:** Thanks. We have removed the entire block formerly at Lines 286–296 (initial manuscript) and redistributed its content to the appropriate sections of *Introduction* (Lines 52–55 of the revised manuscript) and *Results* (Lines 246–255, 268–271, and 279–314 of the revised manuscript).

Line 286-288: Move to INTRO.

**Response:** Thanks. We have moved this sentence to the *Introduction*. For more details, please refer to Lines 52–55 of the revised manuscript.

Line 288-289: Move to Results part.

**Response:** Thanks. We have moved this sentence to the *Results* (3.1 Fungal and bacterial necromass contribution to SOC in agricultural and natural ecosystems). For more details, please refer to Lines 253–255 of the revised manuscript.

*Line 289-296: Most are repeated Results context. Delate.*

**Response:** Thanks. We have deleted these sentences.

Line 298-299: the subheading is many times repeated in this paper. May use something different, may be like "Fungal necromass Greater contribution to SOC by fungal necromass than by bacterial one."

**Response:** Thanks. We have replaced the subheading with a clearer and grammatically polished title—"Fungal necromass contributes two times more to SOC than bacterial necromass" (Line 317 of the revised manuscript).

*Line 300-302 Should included in INTRO, not repeated here.*

**Response:** Thanks. We have removed these sentences and integrated the relevant content into the *Introduction* (Lines 52–55 of the revised manuscript).

Line 302-305: Avoid repeated statement of result. But you need specify the range of the ratio difference among the samples and between your two sets of ecosystems. It may not be true fungal necromass contribution twice as much as bacterial across samples.

**Response:** Thank you for this insightful comment. We have removed the repeated statement of the results in the Discussion and have now explicitly specified the range (along with mean  $\pm$  SE) for both FNC/SOC and BNC/SOC in agricultural and natural ecosystems, as suggested.

**The revised text now reads:**

Our results show that in agricultural ecosystems, FNC/SOC ranged from 0.09% to 97.53% (mean  $\pm$  SE: 34.39  $\pm$  0.67%), and BNC/SOC ranged from 0.81% to 65.00%

(15.65  $\pm$  0.33%). In natural ecosystems, FNC/SOC ranged from 0.92% to 96.29% (29.24  $\pm$  0.51%), and BNC/SOC ranged from 0.25% to 89.45% (14.02  $\pm$  0.36%) (Table 1). The FNC/BNC ratio ranged from 0.02 to 12.74 (2.61  $\pm$  0.06) in agricultural ecosystems and from 0.12 to 44.24 (3.22  $\pm$  0.11) in natural ecosystems (Table 1). Despite substantial variability at the individual sample level, the mean contribution of FNC was approximately twice that of BNC in both ecosystem types. Moreover, the mean FNC/BNC ratio was significantly higher in natural ecosystems than in agricultural ecosystems (P < 0.05; Figure 2).

For further details, please refer to Lines 318–327 of the revised manuscript.

Line 305: if this sentence correct, then what is your study's novelty? If the following discussion about the factors are new, then you may say "The similar variance feature been reported in previous studies, but the reasons unknown. In this study......

**Response:** Thanks for this constructive comment. We acknowledge that the greater contribution of fungal necromass carbon (FNC) to soil organic carbon (SOC) compared to bacterial necromass carbon (BNC) has been reported in previous studies (e.g., Liang et al., 2019; Wang et al., 2021a). However, the novelty of our work lies not in reaffirming this general pattern, but in uncovering systematic differences in the magnitude and drivers of FNC and BNC contributions between agricultural and natural ecosystems at a global scale—a comparison that has not been comprehensively explored until now.

Specifically, although prior research has documented the dominance of FNC in certain ecosystems (e.g., forests, grasslands, or croplands), our study is the first to explicitly compare agricultural and natural ecosystems worldwide and demonstrate that:

- (1) The absolute contributions of both FNC and BNC to SOC are significantly higher in agricultural ecosystems—even though the FNC/BNC ratio is lower—a novel finding that challenges the assumption that natural systems invariably accumulate more microbial necromass;
- (2) The FNC/BNC ratio is significantly lower in agricultural ecosystems than in natural systems, underscoring the impact of management practices on reducing fungal dominance;
- (3) The key drivers of necromass accumulation and composition differ between ecosystem types and between the FNC/BNC ratio and its constituent contributions. Specifically, soil properties (particularly C/N ratio and clay content) govern the contributions of FNC and BNC to SOC in both ecosystems, whereas geographical factors—especially elevation—emerge as the primary drivers of the FNC/BNC ratio. This latter insight is especially novel and underscores the importance of large-scale environmental gradients in regulating microbial residue composition.

We have revised the relevant sentence in the manuscript (Line 305 of the initial manuscript) to better emphasize these novel contributions. The text now reads:

"Although this general pattern has been reported in previous studies (Liang et al., 2019; Wang et al., 2021a; Zhang et al., 2023; Ding et al., 2024), the systematic differences in the magnitude of these contributions between agricultural and natural ecosystems—and their underlying drivers—have remained poorly understood. Our study not only confirms the broad pattern but also elucidates these ecosystem-level disparities and their environmental determinants."

For further details, please refer to Lines 327–332 of the revised manuscript.

Line 308-319: Unfortunately, the discussion are weak, just using some knowledge from publications not with your own analysis or statistical attribution.

**Response:** Thank you for this insightful comment. We have thoroughly revised the Discussion to better integrate our findings and statistical outputs—such as those from Boosted Regression Trees (BRT) and Structural Equation Modeling (SEM)—into the mechanistic interpretation.

**The revised text now reads:**

Consistent with our finding that the contribution of fungal necromass carbon (FNC) to SOC exceeded that of bacterial necromass carbon (BNC) in both ecosystem types (Table 1), the predominance of fungal necromass may be attributed to its more recalcitrant cell wall composition (e.g., chitin) and slower decomposition rate (Wang et al., 2021a). Our BRT and SEM analyses further identified soil clay content and C/N ratio as key drivers of FNC accumulation (Figs. 4a, 5a), reinforcing the importance of organo-mineral associations in the stabilization of fungal-derived carbon.

Line 320-323: If the finding is new, you may rewrite like: In this study we found higher microbial necromss contribution in agricultural system than in natural ecosystems.

For further details, please refer to Lines 333–339 of the revised manuscript.

**Response:** Thanks for the constructive comment. We have revised the sentence to more clearly highlight our novel finding. The text now reads:

Furthermore, our study reveals previously unreported disparities between ecosystem types: the contributions of both fungal and bacterial necromass carbon (FNC and BNC) to SOC were significantly higher in agricultural ecosystems, while the FNC/BNC ratio was substantially elevated in natural ecosystems.

For further details, please refer to Lines 340–343 of the revised manuscript.

Line 324-326: You could use this reason for lower contribution in natural ecosystem but not ending with "potentially resulting in a greater proportion of microbially derived C within SOC (Angst et al., 2021).".

**Response:** Thank you for pointing this out. We have removed the inconsistent content and revised the text to align the reasoning with our data and established conceptual frameworks. The revised sentence now reads:

First, natural ecosystems typically receive larger and more heterogeneous plant-derived carbon inputs than agricultural systems. These inputs expand the plant-derived SOC pool and can dilute the relative contribution of microbial necromass to SOC, thereby resulting in a lower perceived contribution of microbial necromass in natural ecosystems (Angst et al., 2021; Kleber et al., 2021).

For further details, please refer to Lines 345–349 of the revised manuscript.

Line 328-337: The second reason for higher microbial necromass contribution pointed to high quality substrates in agricultural systems, with lower C/N ratio generally. Could you use a correlation respectively of these necromass contribution values to the soils C/N ratio? Lower C/N ratio in agricultural soils is driven by the N fertilization, not necessarily by high quality substrate like legume residue. In fact, agricultural residues are often high C/N ratio, for example wheat straw is over 30.

**Response:** Thank you for this important clarification. We have revised the text to focus on soil C/N ratio rather than "high-quality residues" and have included the requested correlation analyses. The revised sentence now reads:

Second, the significantly lower soil C/N ratio in agricultural ecosystems (10.78) compared to natural ecosystems (27.44) reflects relative nitrogen enrichment, largely resulting from anthropogenic fertilization (Castellano et al., 2015; Chen et al., 2020). This nitrogen-rich environment can enhance microbial carbon use efficiency and alleviate nutrient limitation, thereby promoting the production and accumulation of microbial necromass (Liang et al., 2017). Supporting this mechanism, we found that the contributions of both FNC and BNC to SOC decreased significantly with increasing soil C/N ratio in both agricultural ecosystems (FNC/SOC: R = -0.27, P < 0.001; BNC/SOC: R = -0.29, P < 0.001) and natural ecosystems (FNC/SOC: R = -0.17, P < 0.001; BNC/SOC: R = -0.35, P < 0.001; Figures S6g, S7g). These results further underscore that a lower soil C/N ratio—often indicative of higher nitrogen availability—is a key driver of microbial necromass accumulation. It should be noted that although in situ plant residues in agricultural systems (e.g., cereal straw) may have high C/N ratios, the overall soil C/N ratio is reduced by management practices such as mineral fertilization and the incorporation of low C/N organic amendments.

For further details, please refer to Lines 350–364 of the revised manuscript.

Line 340-342: may be not the difference between the two microbial groups but the difference in microbial behavior between the two systems, which you mentioned later.

**Response:** Thank you for this insightful comment. We fully agree that the observed differences are better explained by the contrasting environmental conditions between agricultural and natural ecosystems—which shape microbial community composition and activity—rather than implying an intrinsic or fixed preference of microbial functional groups.

Accordingly, we have revised the relevant sentence (Lines 365–371 of the revised

manuscript) to clarify that nutrient-rich conditions in agricultural systems typically select for bacterial-dominated communities, whereas resource-heterogeneous environments in natural systems favor fungal dominance. The revised text now reads: "Furthermore, nutrient-rich conditions prevalent in agricultural systems (e.g., due to fertilization) often select for bacterial-dominated communities, as many bacteria exhibit *r*-strategist traits that support rapid growth under high resource availability. In contrast, natural ecosystems—characterized by lower nutrient availability and greater resource heterogeneity—tend to favor fungal dominance, since fungi often function as *K*-strategists with higher efficiency in decomposing complex organic matter under resource-limited conditions (Strickland & Rousk, 2010; Yu et al., 2022)."

*Line 344-349: These are very weak nor robust.*

**Response:** Thanks. We agree that these statements were weak and did not add robust support to the argument. We have therefore removed these statements, and integrated these points into the preceding arguments. For more details, please refer to Lines 365–374 of the revised manuscript.

Line 351-359: These are not sound knowledge. Should link the ratio difference to the difference in SOM accumulation between natural and agricultural systems.

**Response:** Thank you for this insightful comment. We fully agree that merely reporting differences in the FNC/BNC ratio is inadequate without linking it mechanistically to the distinct pathways of soil organic matter (SOM) accumulation across ecosystems. We have thoroughly revised this section (Lines 374–394 of the revised manuscript) to provide a more robust and theory-driven explanation, as detailed below:

A high FNC/BNC ratio signifies a fungal-dominated decomposition pathway. Fungal necromass—rich in recalcitrant compounds such as chitin—is more resistant to decay, and fungal hyphae play a key role in the formation of stable soil aggregates that physically protect organic matter from degradation (Lenardon et al., 2007). This pathway promotes the formation of stable, long-turnover SOC pools essential for long-term carbon sequestration (Six et al., 2006; Lehmann et al., 2020). Furthermore, fungi generally exhibit higher carbon use efficiency than bacteria, meaning a larger proportion of assimilated carbon is allocated to biomass production (and subsequently necromass) rather than being respired as CO2 (Wang & Kuzyakov, 2024). Thus, the fungal-driven pathway characteristic of natural ecosystems represents a highly efficient conversion of plant litter into persistent soil organic matter (Kallenbach et al., 2016; Malik et al., 2016).

Conversely, the lower FNC/BNC ratio observed in agricultural ecosystems reflects a bacterial-dominated pathway, accelerated by practices such as tillage and nutrient amendments. This pathway is associated with faster carbon cycling and greater carbon loss through respiration. Although microbial necromass can accumulate under these conditions—sometimes contributing more significantly to a reduced total SOC pool—the resulting carbon is often less stabilized (Zhou et al., 2023).

Therefore, the FNC/BNC ratio serves not merely as a descriptive metric, but as a functional biomarker that elucidates fundamental differences in the stability and persistence of SOM between managed agricultural systems and natural ecosystems.

The contents in 4.2 should be sued in discussion part 4.1. When the reason of the changes is in discussion, you present these results from statistics to support or to cohere your finding. Not presented separately while leaving your discussion often pale.

**Response:** We greatly appreciate the reviewer's comment, which has helped strengthen the integration of results and discussion. We have carefully addressed this suggestion in our revision in the following ways:

First, in direct response to this comment, we have deeply integrated the discussion of key drivers into Section 4.1 (Fungal necromass contributes two times more to SOC than bacterial necromass). Mechanistic explanations for core patterns—such as the dominant roles of soil C/N and clay content in governing the contributions of FNC and BNC to SOC, and the negative correlations between these contributions and the soil C/N ratio (Lines 337–339, 350–361 of the revised manuscript)—are now presented alongside the corresponding results. This ensures that each major finding is immediately accompanied by its interpretive context.

Second, we retained Section 4.2 (Driving factors of the change in fungal and bacterial necromass contribution to SOC and their ratio; Lines 414–456) in accordance with the reviewer's earlier feedback. The reviewer rightly pointed out that the analysis of driving factors, being a synthesis derived from disparate studies without controlled treatments, "should be presented in the Discussion part, not the direct results presented here." We interpreted this as a directive to relocate the synthesis on drivers from the Results to the Discussion. Thus, Section 4.2 now serves as a dedicated integrative discussion that compares the relative importance of all factor types—geographical, climatic, soil physicochemical, and biotic—across the different response variables, rather than merely repeating results.

We believe this revised structure—incorporating immediate mechanistic interpretation within Section 4.1, followed by a synthesized discussion of drivers in Section 4.2—best addresses both of the reviewer's comments. It achieves seamless integration of results and interpretation while providing a appropriate space for the synthetic analysis that rightly belongs in the Discussion.

We hope this revised organization meets with the reviewer's approval.

Subheading 4.3, statement about limitations are honest. But need to mention that sampling conditions may not be comparable so as to the large variability.

**Response:** Thank you for raising this critical point. As suggested, we have now

explicitly acknowledged this limitation in the revised manuscript. The specific statement reads:

Furthermore, the compiled studies employed varied methodologies regarding sampling time, depth, and laboratory protocols. While such heterogeneity is an inherent challenge in global meta-analyses, it likely introduces additional variability and may constrain the direct comparability of certain data points.

For further details, please refer to Lines 470–473 of the revised manuscript.

**Conclusions**

Line 429-430: why not "FNC two times as much as BNC."

**Response:** Thank you for this insightful comment. We have revised the sentence to now read:

"Our results indicate that, on average, fungal necromass carbon (FNC) contributes approximately twice as much to soil organic carbon (SOC) as bacterial necromass carbon (BNC) in both agricultural and natural ecosystems."

For further details, please refer to Lines 483–485 of the revised manuscript.

*Line 432-434: significantly but slightly.*

**Response:** Thank you for your meticulous and professional suggestion. We have revised the sentence as follows:

"The FNC/BNC ratio was significantly higher in natural ecosystems than in agricultural ecosystems, albeit with a modest effect size, and was primarily driven by geographical factors—particularly elevation."

For further details, please refer to Lines 487–490 of the revised manuscript.

Line 434-437: no evidence of "consistent trends", as for the large variability.

**Response:** Thank you for highlighting this important point. We have revised the concluding statement to more accurately reflect the statistically significant differences observed between ecosystem types, rather than implying uniform trends across all sites. The revised text now reads:

"Our findings demonstrate that, despite considerable variability among individual sampling sites, statistically significant differences exist between agricultural and natural ecosystems in the contributions of fungal and bacterial necromass carbon (FNC and BNC) to soil organic carbon (SOC), as well as in the FNC/BNC ratio, at a global scale. These results underscore a potential fundamental divergence in the pathways and mechanisms of carbon turnover and stabilization between these two broad ecosystem types."

For further details, please refer to Lines 490–496 of the revised manuscript.

Line 437-440: Mention about added value of your study compared to previous study, or future perspectives.

**Response:** Thank you for highlighting this valuable suggestion. We have revised the content as follows:

"These insights provide novel evidence that ecosystem management type (agricultural versus natural) is a key determinant of the pathways through which microbial necromass contributes to the global soil organic carbon (SOC) pool. Future studies that integrate microbial community composition with necromass dynamics across a broader range of biomes will be essential to predict ecosystem-specific responses of this critical carbon pool to global change."

For further details, please refer to Lines 496–502 of the revised manuscript.

**References**

- Anderson, T.H., Domsch, K.H., 1989. Ratios of microbial biomass carbon to total organic carbon in arable soils. Soil Biol. Biochem. 21, 471–479. https://doi.org/10.1016/0038-0717(89)90117-X
- Angst, G., Mueller, K.E., Nierop, K.G., Simpson, M.J., 2021. Plant-or microbial-derived? A review on the molecular composition of stabilized soil organic matter. Soil Biol. Biochem. 156, 108189. https://doi.org/10.1016/j.soilbio.2021.108189
- Chen, X., Hu, Y., Xia, Y., Zheng, S., Ma, C., Rui, Y., He, H., Huang, D., Zhang, Z., Ge, T., Wu, J., Guggenberger, G., Kuzyakov, Y., Su, Y., 2021. Contrasting pathways of carbon sequestration in paddy and upland soils. Global Change Biol. 27, 2478–2490. https://doi.org/10.1111/gcb.15595
- Clocchiatti, A., Hannula, S.E., Hundscheid, M.P., Klein Gunnewiek, P.J., de Boer, W., 2021. Stimulated saprotrophic fungi in arable soil extend their activity to the rhizosphere and root microbiomes of crop seedlings. Environ. Microbiol. 23, 6056–6073. https://doi.org/10.1111/1462-2920.15563
- Cotrufo, M.F., Wallenstein, M.D., Boot, C.M., Denef, K., Paul, E., 2013. The Microbial Efficiency-Matrix Stabilization (MEMS) framework integrates plant litter decomposition with soil organic matter stabilization: do labile plant inputs form stable soil organic matter? Global Change Biol. 19, 988–995. https://doi.org/10.1111/gcb.12113
- de Boer, W.D., Folman, L.B., Summerbell, R.C., Boddy, L., 2005. Living in a fungal world: impact of fungi on soil bacterial niche development. FEMS Microbiol. Rev. 29, 795–811. https://doi.org/10.1016/j.femsre.2004.11.005
- Ding, Z., Mou, Z., Li, Y., Liang, C., Xie, Z., Wang, J., Hui, D., Lambers, H., Sardans, J., Peñuelas, J., Xu, H., Liu, Z., 2024. Spatial variation and controls of soil microbial necromass carbon in a tropical montane rainforest. STOTEN. 921, 170986. https://doi.org/10.1016/j.scitotenv.2024.170986
- Gao, W., Duan, X., Chen, X., Wei, L., Wang, S., Wu, J., Zhu, Z., 2024. Iron-carbon complex types and bonding forms jointly control organic carbon mineralization in paddy soils. STOTEN. 953, 176117. https://doi.org/10.1016/j.scitotenv.2024.176117
- Ghezzehei, T.A., Sulman, B., Arnold, C.L., Bogie, N.A., Berhe, A.A., 2019. On the role of soil water retention characteristic on aerobic microbial respiration.

- BIOGEOSCIENCES. 16, 1187–1209. https://doi.org/10.5194/bg-16-1187-2019
- Han, B., Yao, Y., Wang, Y., Su, X., Ma, L., Chen, X., Li, Z., 2024. Microbial traits dictate soil necromass accumulation coefficient: A global synthesis. Global Change Biol. 33, 151–161. https://doi.org/10.1111/geb.13776
- Hättenschwiler, S., Tiunov, A.V., Scheu, S., 2005. Biodiversity and litter decomposition in terrestrial ecosystems. Annu. Rev. Ecol. Evol. Syst. 36, 191–218. https://doi.org/10.1146/annurev.ecolsys.36.112904.151932
- Irving, T.B., Alptekin, B., Kleven, B., Ané, J.M., 2021. A critical review of 25 years of glomalin research: a better mechanical understanding and robust quantification techniques are required. New Phytol. 232, 1572–1581. https://doi.org/10.1111/nph.17713
- Kleber, M., Bourg, I.C., Coward, E.K., Hansel, C.M., Myneni, S.C., Nunan, N., 2021. Dynamic interactions at the mineral—organic matter interface. NAT REV EARTH ENV. 2, 402–421. https://doi.org/10.1038/s43017-021-00162-y
- Klink, S., Keller, A.B., Wild, A.J., Baumert, V.L., Gube, M., Lehndorff, E., Meyer, N., Mueller, C.W., Phillips, R.P., Pausch, J., 2022. Stable isotopes reveal that fungal residues contribute more to mineral-associated organic matter pools than plant residues. Soil Biol. Biochem. 168, 108634. https://doi.org/10.1016/j.soilbio.2022.108634
- Liang, C., Amelung, W., Lehmann, J., Kästner, M., 2019. Quantitative assessment of microbial necromass contribution to soil organic matter. Global Change Biol. 25, 3578–3590. https://doi.org/10.1111/gcb.14781
- Liu, C., Tian, J., Cheng, K., Xu, X., Wang, Y., Liu, X., Liu, Z., Bian, R., Zhang, X., Xia, S., Zheng, J., Li, L., Pan, G., 2023. Topsoil microbial biomass carbon pool and the microbial quotient under distinct land-use types across China: A data synthesis. SSE. 2, 5. https://doi.org/10.48130/SSE-2023-0005
- Qiu, H., Zheng, X., Ge, T., Dorodnikov, M., Chen, X., Hu, Y., Kuzyakov, Y., Wu, J., Su, Y., Zhang, Z., 2017. Weaker priming and mineralisation of low molecular weight organic substances in paddy than in upland soil. Eur. J. Soil Biol. 83, 9–17. https://doi.org/10.1016/j.ejsobi.2017.09.008
- Rillig, M.C., 2004. Arbuscular mycorrhizae, glomalin, and soil aggregation. Can. J. Soil Sci. 84, 355–363. https://doi.org/10.4141/S04-003
- Shi, W., Gao, D., Zhang, Z., Ding, J., Zhao, C., Wang, H., Hagedorn, F., 2024. Exploring global data sets to detect changes in soil microbial carbon and nitrogen over three decades. EARTHS FUTURE 12, e2024EF004733. https://doi.org/10.1029/2024EF004733x
- Štursová, M., Žifčáková, L., Leigh, M.B., Burgess, R., Baldrian, P., 2012. Cellulose utilization in forest litter and soil: identification of bacterial and fungal decomposers. FEMS Microbiol. Ecol. 80, 735–746. https://doi.org/10.1111/j.1574-6941.2012.01343.x
- Von Haden, A.C., Yang, W.H., DeLucia, E.H., 2020. Soils' dirty little secret: Depth-based comparisons can be inadequate for quantifying changes in soil organic carbon and other mineral soil properties. Global Change Biol. 26, 3759–3770. https://doi.org/10.1111/gcb.15124

- Wang, B., An, S., Liang, C., Liu, Y., Kuzyakov, Y., 2021a. Microbial necromass as the source of soil organic carbon in global ecosystems. Soil Biol. Biochem. 162, 108422. https://doi.org/10.1016/j.soilbio.2021.108422
- Wang, Z., Zhao, M., Yan, Z., Yang, Y., Niklas, K.J., Huang, H., Mipam, T.D., He, X., Hu, H., Wright, S.J., 2022. Global patterns and predictors of soil microbial biomass carbon, nitrogen, and phosphorus in terrestrial ecosystems. Catena 211, 106037. https://doi.org/10.1016/j.catena.2022.106037
- Wendt, J.W., Hauser, S., 2013. An equivalent soil mass procedure for monitoring soil organic carbon in multiple soil layers. Eur. J. Soil Sci. 64, 58–65. https://doi.org/10.1111/ejss.12002
- Xu, S., Song, X., Zeng, H., Wang, J., 2024. Soil microbial necromass carbon in forests: A global synthesis of patterns and controlling factors. Soil Ecol. Lett. 6(4), 240237. https://doi.org/10.1007/s42832-024-0237-3
- Yao, Y., Han, B., Dong, X., Zhong, Y., Niu, S., Chen, X., Li, Z., 2024. Disentangling the variability of symbiotic nitrogen fixation rate and the controlling factors. Global Change Biol. 30, e17206. https://doi.org/10.1111/gcb.17206
- Zhang, Q., Li, X., Liu, J., Liu, J., Han, L., Wang, X., Liu, H., Xu, M., Yang, G., Ren, C., Han, X., 2023. The contribution of microbial necromass carbon to soil organic carbon in soil aggregates. Appl. Soil Ecol. 190, 104985. https://doi.org/10.1016/j.apsoil.2023.104985
- Zhang, X., Jia, J., Chen, L., Chu, H., He, J.S., Zhang, Y., Feng, X., 2021. Aridity and NPP constrain contribution of microbial necromass to soil organic carbon in the Qinghai-Tibet alpine grasslands. Soil Biol. Biochem. 156, 108213. https://doi.org/10.1016/j.soilbio.2021.108213
- Zhou, R., Liu, Y., Dungait, J.A., Kumar, A., Wang, J., Tiemann, L.K., Zhang, F., Kuzyakov, Y., Tian, J., 2023. Microbial necromass in cropland soils: A global meta analysis of management effects. Global Change Biol. 29, 1998–2014. https://doi.org/10.1111/gcb.16613

---

## Author Comment (AC2)

Dear Editor,

We would like to thank you, and the reviewers for the contributions to this manuscript.

The constructive feedback has been extremely helpful. We have accepted all the

changes suggested and made the appropriate changes to the study. We believe that the

manuscript is considerably clearer and more impactful as a result.

Attached please find our point-by-point responses to the reviewer's comments.

We thank you for your consideration and hope you will find this version suitable for

publication in Earth System Science Data.

Best regards,

Zhiqiang Wang, and on behalf of all co-authors

Sichuan Zoige Alpine Wetland Ecosystem National Observation and Research Station,

Southwest Minzu University

Chengdu, 610041, PR China

E-mail: wangzq@swun.edu.cn

**Response to reviewer's comments**

**Responses to the Reviewer's comments**

The paper presents an excellent and timely study, offering a comprehensive global-scale analysis of the contributions of fungal and bacterial necromass carbon (FNC and BNC) to soil organic carbon (SOC) across agricultural and natural ecosystems. The manuscript is well-written, methodologically rigorous, and addresses a topic of significant importance in soil biogeochemistry. The findings provide valuable insights into microbial-mediated carbon stabilization mechanisms in terrestrial ecosystems. I suggest this study highly suitable for publication in ESSD, however, some questions should be resolved before final acceptance.

**Response:** We sincerely thank the reviewer for the positive and encouraging comments on our manuscript. We particularly appreciate the reviewer's recognition of the global-scale analysis, methodological rigor, and significance of our study to the field of soil biogeochemistry.

We have carefully considered all the points raised by the reviewer. In the sections below, we provide a point-by-point response to the specific questions and have revised the manuscript accordingly to address them. We believe that these revisions have further strengthened the quality and clarity of our work.

**Major concerns**

In Section 2.1, The authors should justify the use of interpolated data (e.g., for MAT, MAP, and soil properties) obtained from public databases. Please address the potential uncertainties and describe any steps taken to validate these values against site-specific conditions or to quantify the associated error in the analysis.

**Response:** Thanks for this important suggestion. We agree that acknowledging and addressing the potential uncertainties associated with these datasets is crucial for the robustness of our global-scale analysis. Below, we provide a justification for their use and describe the steps we took to mitigate potential issues.

The primary rationale for employing globally interpolated datasets (e.g., WorldClim, SoilGrids) was to ensure consistent, continuous, and spatially complete coverage of environmental variables across all 486 globally distributed sites. The original publications from which microbial necromass data were extracted frequently did not report the full suite of climatic and soil variables required for our unified analysis. By using these standardized, high-resolution global datasets, we maintained methodological consistency and mitigated potential biases arising from missing data.

We acknowledge that interpolated data inherently contain uncertainties. To address this,

we took the following steps:

- (1) We exclusively used globally recognized and widely cited databases (e.g., WorldClim v2.1 with a 30-arc second resolution, SoilGrids 2.0), which represent the current state-of-the-art in global spatial interpolation and are extensively used in global ecological and biogeochemical studies (e.g., Lu et al., 2022; Ren et al., 2024; Shi et al., 2025; Zhou et al., 2025).
- (2) After retrieving missing value from gridded data, we typically calibrate them against field-reported values via a field-anchored bias correction (i.e., a site- or region-specific "delta" adjustment) to minimize errors introduced by gridded data.
- (3) Our statistical approach inherently accounts for data uncertainty. The performance metrics of our models (e.g., the R² values ranging from 23% to 66% in our Boosted Regression Tree analysis, as shown in Figure 4) already reflect the unexplained variance, which partly incorporates the measurement and interpolation errors of all input variables. The fact that we still identified strong and significant drivers suggests that the signals we detected are robust enough to overcome the background noise, including potential errors from interpolation.

In response to the reviewer's comment, we have revised the Section 2.1 (Data collection) in the revised manuscript to explicitly address this point. The added text reads:

"We supplemented missing climatic and soil variables using high-resolution, globally interpolated datasets to ensure consistent spatial coverage across all sites. After retrieving missing value from gridded data, we typically calibrate them against field-reported values via a field-anchored bias correction (i.e., a site- or region-specific "delta" adjustment) to minimize errors introduced by gridded data. While the use of such data introduces inherent uncertainties, these databases are widely adopted in global-scale ecological analyses and provide the most feasible approach for a unified assessment." For further detail, please see Lines 156–163 of the revised manuscript.

Section 3.2 presents a highly detailed and, at times, repetitive description of the results. This level of minutia can obscure the key findings for the reader. To improve clarity and impact, I strongly recommend that the authors streamline this section. The text should be condensed to focus on the primary results, avoiding a minute description of every statistical outcome. Reorganizing the content into clearer thematic paragraphs would also significantly enhance its readability.

**Response:** Thank you for this constructive comment. As recommended, we have reorganized the Section 3.2 as followings:

"Soil physicochemical factors were the most important influence on the contributions of FNC and BNC to SOC across both ecosystem types (Figures 3a–d, 4a–d). Specifically, they explained 16% and 17% of the variance in the contributions of FNC and BNC to SOC in agricultural ecosystems, respectively (Figures 3a, c), and 20% and 24% in natural ecosystems (Figures 3b, d). BRTs corroborated this pattern, with soil physicochemical factors showing the highest relative influence (51% for FNC, and 44%

for BNC) in agricultural systems and 44% in natural systems (Figures 4a–d). All BRT models were significant (P < 0.001), with explained variance 36–66%. While soil factors dominated overall, responses to individual variables differed between ecosystems. In detail, in agricultural systems, the C/N ratio ranked third for FNC after clay and SOC (Figure 4a), whereas C/N was the top predictor for FNC in natural systems and for BNC in both ecosystems (Figures 4b–d). Consistently, linear models showed declines in the contributions of FNC and BNC with increasing C/N in both ecosystems (Figures S5g, S6g). SEMs yielded convergent results, indicating both direct and indirect pathways (Figures 5a–d, 6a–d). Notably, the direct and total effects of soil physicochemical factors on FNC were negative in agricultural but positive in natural ecosystems (Figures 5a, b, 6a, b), whereas the effects on BNC were negative in both ecosystem types (Figures 5c, d, 6c, d).

Our results indicated that geographical factors were the most important contributors to explain the FNC/BNC ratio in both agricultural and natural ecosystems, accounting for 21% and 10% of the explained variance in the FNC/BNC ratio, respectively (Figures 3e, f). The results of the BRTs suggested that geographical factors played a similar role in explaining the FNC/BNC ratio (Figures 4e, f). In the BRT models, geographical factors emerged as the primary influencers of the FNC/BNC ratio in agricultural and natural ecosystems, accounting for 32% and 44% of the variance in each case, respectively (Figures 4e, f). To be more specific, elevation was the most significant geographical factors influencing the FNC/BNC ratio in both ecosystems (Figures 4e, f). Moreover, the FNC/BNC ratio in agricultural and natural ecosystems show significantly increased with an increase elevation (Figure S7a). The results of SEMs also indicated that geographical factors were the most influential factors for the FNC/BNC ratio in agricultural and natural ecosystems, exerting both direct and indirect effects on this ratio (Figures 5e, 6e), with the standardized total effect being positive (Figures 5f, 6f)."

For further details, please see Lines 235–267 of the revised manuscript.

Meanwhile, I suggest the authors separately describe the effects of driving factors on the contributions with agricultural and natural ecosystems. Also, in the section 4.2, the authors should better discuss it separately about agricultural and natural ecosystems.

**Response:** Thanks for the constructive comment. As recommended, we have separately described the effects of driving factors on the contributions with agricultural and natural ecosystems as followings:

"While soil factors dominated overall, responses to individual variables differed between ecosystems. In detail, in agricultural systems, the C/N ratio ranked third for FNC after clay and SOC (Figure 4a), whereas C/N was the top predictor for FNC in natural systems and for BNC in both ecosystems (Figures 4b–d). Consistently, linear models showed declines in the contributions of FNC and BNC with increasing C/N in both ecosystems (Figures S5g, S6g). SEMs yielded convergent results, indicating both direct and indirect pathways (Figures 5a–d, 6a–d). Notably, the direct and total effects of soil physicochemical factors on FNC were negative in agricultural but positive in

natural ecosystems (Figures 5a, b, 6a, b), whereas the effects on BNC were negative in both ecosystem types (Figures 5c, d, 6c, d).".

For further details, please see Lines 242–252 of the revised manuscript.

In addition, we have reorganized and discussed it separately about agricultural and natural ecosystems in the Section 4.2. The revised text now reads:

"In agricultural ecosystems, high soil N levels primarily result from fertilization (Chen et al., 2020). In contrast, natural ecosystems experience minimal anthropogenic disturbance, N often acts as the key limiting factor for microbial activity (Elser et al., 2007). Under N-limited conditions, microbes (both fungi and bacteria) allocate more energy and C resources to the synthesis of N-acquiring enzymes (e.g., proteases and chitinases). This shift in metabolic strategy reduces the C allocated to biomass synthesis, thereby diminishing the amount of C ultimately converted into microbial necromass (Mooshammer et al., 2014; Liu et al., 2024). Thus, although microbial community composition differs between natural and agricultural ecosystems, the regulatory role of soil C/N ratio in shaping their structure and function remains consistent (Han et al., 2024). In our study, soil clay content was identified as the predominant factor governing the contribution of FNC to SOC in agricultural ecosystems (Figure 4a), with this contribution increasing concomitantly with clay content (Figure S5d). This suggests that soils with higher clay and silt contents generally accumulate greater amounts of microbial residues, particularly those derived from fungi, which can be attributed to the promotion of stable organo-mineral complex formation by abundant fine soil particles (Six et al., 2006 and Liang et al., 2017). Furthermore, although agricultural management practices often disturb soil structure, they simultaneously enhance clay enrichment and aggregate formation, thereby providing effective physical protection for the long-term stabilization of fungal-derived C (Chen et al., 2020; Mou et al., 2021; Zhou et al., 2023).".

For further details, please see Lines 344–365 of the revised manuscript.

The Discussion would benefit from a sharper focus on the novelty of this study. Currently, the overemphasis on aligning with previous findings (e.g., Lines 305–306, 340–341) detracts from highlighting the new insights. This is apparent in Section 4.1, where the interpretation of results, such as the elevated FNC and BNC in agricultural ecosystems, needs more mechanistic depth. The authors should use their own analytical evidence (e.g., from BRT and SEM on C/N ratio and clay content) to explain these patterns, rather than merely stating them. The discussion should use prior literature to frame the study's unique conclusions, not just to confirm them.

**Response:** Following the constructive comment, we have reorganized and revised some parts of Section 4.1 in the manuscript. The updated text now reads:

"Although this general pattern has been reported in previous studies (Liang et al., 2019; Wang et al., 2021a; Zhang et al., 2023; Ding et al., 2024), the systematic differences in the magnitude of these contributions between agricultural and natural ecosystems—and their underlying drivers—have remained poorly understood. Our study not only

confirms the broad pattern but also elucidates these ecosystem-level disparities and their environmental determinants. Consistent with our finding that the contribution of fungal necromass carbon (FNC) to SOC exceeded that of bacterial necromass carbon (BNC) in both ecosystem types (Table 1), the predominance of fungal necromass may be attributed to its more recalcitrant cell wall composition (e.g., chitin) and slower decomposition rate (Wang et al., 2021a). Our BRT and SEM analyses further identified soil clay content and C/N ratio as key drivers of FNC accumulation (Figs. 4a, 5a), reinforcing the importance of organo-mineral associations in the stabilization of fungal-derived carbon."

For further details, please refer to Lines 275–287 of the revised manuscript.

Furthermore, we have reorganized and revised the specific paragraph (contained the content in Lines 340–341 of the original manuscript), strengthening the support for our findings by integrating relevant pre-existing literature. The updated text now reads: "Furthermore, nutrient-rich conditions prevalent in agricultural systems (e.g., due to fertilization) often select for bacterial-dominated communities, as many bacteria exhibit r-strategist traits that support rapid growth under high resource availability. In contrast, natural ecosystems—characterized by lower nutrient availability and greater resource heterogeneity—tend to favor fungal dominance, since fungi often function as Kstrategists with higher efficiency in decomposing complex organic matter under resource-limited conditions (Strickland & Rousk, 2010; Yu et al., 2022). This shift in microbial community composition is reflected in our results, which show a significantly higher FNC/BNC ratio in natural ecosystems across our global dataset (Figure 2c, Table 1). A high FNC/BNC ratio signifies a fungal-dominated decomposition pathway. Fungal necromass—rich in recalcitrant compounds such as chitin—is more resistant to decay, and fungal hyphae play a key role in the formation of stable soil aggregates that physically protect organic matter from degradation (Lenardon et al., 2007). This pathway promotes the formation of stable, long-turnover SOC pools essential for longterm carbon sequestration (Six et al., 2006; Lehmann et al., 2020). Furthermore, fungi generally exhibit higher carbon use efficiency than bacteria, meaning a larger proportion of assimilated carbon is allocated to biomass production (and subsequently necromass) rather than being respired as CO2 (Wang & Kuzyakov, 2024). Thus, the fungal-driven pathway characteristic of natural ecosystems represents a highly efficient conversion of plant litter into persistent soil organic matter (Kallenbach et al., 2016; Malik et al., 2016). Conversely, the lower FNC/BNC ratio observed in agricultural ecosystems reflects a bacterial-dominated pathway, accelerated by practices such as tillage and nutrient amendments. This pathway is associated with faster carbon cycling and greater carbon loss through respiration. Although microbial necromass can accumulate under these conditions—sometimes contributing more significantly to a reduced total SOC pool—the resulting carbon is often less stabilized (Zhou et al., 2023). Therefore, the FNC/BNC ratio serves not merely as a descriptive metric, but as a functional biomarker that elucidates fundamental differences in the stability and persistence of SOM between managed agricultural systems and natural ecosystems.". For further details, please refer to Lines 306–335 of the revised manuscript.

Minor concerns

Line 21: Delete this sentence.

**Response:** Done.

Lines 78–81: Suggest change into "Previous studies indicated that the contributions of FNC and BNC to SOC depended on the type of ecosystems (Wang et al., 2021a; Cao et al., 2023; Xu et al., 2024)."

**Response:** Thanks. We have rewritten in the revised manuscript (Lines 77–79).

Lines 126–127: Natural ecosystems include grasslands and forests. What habitats does the agricultural ecosystem consist of? Please clarify this carefully.

**Response:** Thanks. We have explicitly classified the agricultural ecosystem into dry land, irrigated cropland, and submerged paddy. For further details, please refer to Lines 123–124 of the revised manuscript.

Lines 182–183: Why is the threshold for the variance inflation factor set at 3.3 instead of the more common 5 or 10 that we commonly used?

**Response:** Thank you for this insightful and important comment. The choice of a more conservative Variance Inflation Factor (VIF) threshold of 3.3, as opposed to the more commonly cited values of 5 or 10, was a deliberate decision to ensure the robustness and reliability of our models by more rigorously minimizing multicollinearity.

The detailed justification for selecting this specific threshold of 3.3 as following:

- (1) Conventional Thresholds and Their Implications. (a) VIF < 10: This is a very lenient standard, more common in earlier statistical applications. It implies that 90% of an independent variable's variance can be explained by the other independent variables (since 1 1/10 = 0.9). In modern research demanding higher model precision, this threshold is often considered too permissive and may fail to effectively eliminate problematic collinearity. (b) VIF < 5: This is a moderate and frequently used standard, deemed acceptable in many fields. It indicates that up to 80% of a variable's variance is explained by others. This threshold often provides a reasonable balance in many situations.
- (2) Rationale for a Stricter Threshold (VIF < 3.3). Our reference to Kock (2015) is pivotal here. This literature advocates for and substantiates the necessity of a stricter VIF threshold, primarily based on the following points: (a) although this study ultimately uses BRTs and SEM, the threshold proposed by Kock (2015) was initially developed within the context of Partial Least Squares Structural Equation Modeling (PLS-SEM) for comprehensive collinearity assessment. This concept has since been adopted by numerous researchers and applied to a wider range of multivariate statistical models as a gold standard for ensuring predictor independence; (b) A variable with a VIF of 5 still has 20% of its variance inflated by other variables in the model. This remains a non-negligible

proportion that can distort regression coefficient estimates, making them unstable and difficult to interpret. Setting the threshold at 3.3 ensures that no more than approximately 30% of any predictor's variance is explained by others  $(1 - 1/3.3 \approx 0.7)$ . This more effectively guarantees that the influence of each predictor on the response variable is relatively independent, leading to more reliable and trustworthy model outcomes; (c) In ecology and environmental sciences, many environmental drivers (e.g., temperature, precipitation, soil nutrients) are inherently correlated. Employing a strict VIF threshold proactively addresses these issues during the variable selection stage. This ensures that the "most important factors" subsequently identified in the Boosted Regression Trees and Structural Equation Models are genuinely influential, not merely appearing significant due to high correlations with other excluded variables. This significantly strengthens the robustness of the study's conclusions.

Therefore, our selection of the threshold value of 3.3 was not an arbitrary choice, but was grounded in established literature and driven by our commitment to more stringent criteria for data integrity and model robustness.

We sincerely hope this clarification adequately addresses your concern.

Lines 230–233: Suggest delete this sentence. Just provide an objective description of the result, without delving into other details.

**Response:** Thanks. We have deleted the sentence, and revised the respective section to provide a more objective description of the results. For further details, please refer to Lines 235–252 of the revised manuscript.

Lines 286–296: This section contains too much overlap with the introduction and results sections. Suggest delete it.

**Response:** Done.

*Lines 300–302: Delete this sentence.*

Response: Done.

**References**

Chen, G., Ma, S., Tian, D., Xiao, W., Jiang, L., Xing, A., Zou, A., Zhou, L., Shen, H., Zheng, C., Ji, C., He, H., Zhu, B., Liu, L., Fang, J., 2020. Patterns and determinants of soil microbial residues from tropical to boreal forests. Soil Biol. Biochem. 151, 108059. https://doi.org/10.1016/j.soilbio.2020.108059

Ding, Z., Mou, Z., Li, Y., Liang, C., Xie, Z., Wang, J., Hui, D., Lambers, H., Sardans, J., Peñuelas, J., Xu, H., Liu, Z., 2024. Spatial variation and controls of soil microbial necromass carbon in a tropical montane rainforest. STOTEN. 921, 170986. https://doi.org/10.1016/j.scitotenv.2024.170986

Elser, J.J., Bracken, M.E., Cleland, E.E., Gruner, D.S., Harpole, W.S., Hillebrand, H., Ngai, J.T., Seabloom, E.W., Shurin, J.B., Smith, J.E., 2007. Global analysis of

- nitrogen and phosphorus limitation of primary producers in freshwater, marine and terrestrial ecosystems. Ecol. Lett. 10, 1135–1142. https://doi.org/10.1111/j.1461-0248.2007.01113.x
- Han, B., Yao, Y., Wang, Y., Su, X., Ma, L., Chen, X., Li, Z., 2024. Microbial traits dictate soil necromass accumulation coefficient: A global synthesis. Global Ecol. Biogeogr. 33(1), 151–161. https://doi.org/10.1111/geb.13776
- Kallenbach, C.M., Frey, S.D., Grandy, A.S., 2016. Direct evidence for microbial-derived soil organic matter formation and its ecophysiological controls. Nat. Commun. 7, 13630. https://doi.org/10.1038/ncomms13630
- Kock, N., 2015. Common method bias in PLS-SEM: A full collinearity assessment approach. International Journal of e-Collaboration (IJeC) 11, 1–10. https://doi.org/10.4018/ijec.2015100101
- Lehmann, J., Hansel, C.M., Kaiser, C., Kleber, M., Maher, K., Manzoni, S., Nunan, N., Reichstein, M., Schimel, J.P., Torn, M.S., Wieder, W.R., Kögel-Knabner, I., 2020. Persistence of soil organic carbon caused by functional complexity. Nat. Geosci. 13, 529–534. https://doi.org/10.1038/s41561-020-0612-3
- Lenardon, M.D., Whitton, R.K., Munro, C.A., Marshall, D., Gow, N.A.R., 2007. Individual chitin synthase enzymes synthesize microfibrils of differing structure at specific locations in the Candida albicans cell wall. Mol. Microbiol. 66, 1164–1173. https://doi.org/10.1111/j.1365-2958.2007.05990.x
- Liang, C., Amelung, W., Lehmann, J., Kästner, M., 2019. Quantitative assessment of microbial necromass contribution to soil organic matter. Global Change Biol. 25, 3578–3590. https://doi.org/10.1111/gcb.14781
- Liang, C., Schimel, J.P., Jastrow, J.D., 2017. The importance of anabolism in microbial control over soil carbon storage. Nat. Microbiol. 2, 17105. https://doi.org/10.1038/nmicrobiol.2017.105
- Liu, X., Tian, Y., Heinzle, J., Salas, E., Kwatcho Kengdo, S., Borken, W., Schindlbacher, A., Wanek, W., 2024. Long term soil warming decreases soil microbial necromass carbon by adversely affecting its production and decomposition. Global Change Biol. 30, e17379. https://doi.org/10.1111/geb.17379
- Lu, J.L., Jia, P., Feng, S.W., Wang, Y.T., Zheng, J., Ou, S.N., Wu, Z.H., Liao, B., Shu, W.S., Liang, J.L., Li, J.T., 2022. Remarkable effects of microbial factors on soil phosphorus bioavailability: a country-scale study. Global Change Biol. 28, 4459–4471. https://doi.org/10.1111/gcb.16213
- Malik, A.A., Chowdhury, S., Schlager, V., Oliver, A., Puissant, J., Vazquez, P.G., Jehmlich, N., von Bergen, M., Griffiths, R., Gleixner, G., 2016. Soil fungal: bacterial ratios are linked to altered carbon cycling. Front. Microbiol. 7, 1247. https://doi.org/10.3389/fmicb.2016.01247
- Mooshammer, M., Wanek, W., Zechmeister-Boltenstern, S., Richter, A., 2014. Stoichiometric imbalances between terrestrial decomposer communities and their resources: mechanisms and implications of microbial adaptations to their resources. Front. Microbiol. 5, 22. https://doi.org/10.3389/fmicb.2014.00022
- Mou, Z., Kuang, L., He, L., Zhang, J., Zhang, X., Hui, D., Li, Y., Wu, W., Mei, Q., He,

- X., Kuang, Y., Wang, J., Wang, Y., Lambers, H., Sardans, J., Peñuelas, J., Liu, Z., 2021. Climatic and edaphic controls over the elevational pattern of microbial necromass in subtropical forests. Catena 207, 105707. https://doi.org/10.1016/j.catena.2021.105707
- Ren, S., Wang, T., Guenet, B., Liu, D., Cao, Y., Ding, J., Smith P., Piao, S., 2024. Projected soil carbon loss with warming in constrained Earth system models. Nat. Commun. 15, 102. https://doi.org/10.1038/s41467-023-44433-2
- Shi, G., Sun, W., Shangguan, W., Wei, Z., Yuan, H., Li, L., Sun, X., Zhang, Y., Liang, H., Li, D., Huang, F., Li, Q., Dai, Y., 2025. A China dataset of soil properties for land surface modelling (version 2, CSDLv2). Earth Syst. Sci. Data 17, 517–543. https://doi.org/10.5194/essd-17-517-2025
- Six, J., Frey, S.D., Thiet, R.K., Batten, K.M., 2006. Bacterial and fungal contributions to carbon sequestration in agroecosystems. Soil Sci. Soc. Am. J. 70, 555–569. https://doi.org/10.2136/sssaj2004.0347
- Strickland, M.S., Rousk, J., 2010. Considering fungal: bacterial dominance in soils—methods, controls, and ecosystem implications. Soil Biol. Biochem. 42, 1385—1395. https://doi.org/10.1016/j.soilbio.2010.05.007
- Wang, B., An, S., Liang, C., Liu, Y., Kuzyakov, Y., 2021a. Microbial necromass as the source of soil organic carbon in global ecosystems. Soil Biol. Biochem. 162, 108422. https://doi.org/10.1016/j.soilbio.2021.108422
- Wang, C., Kuzyakov, Y., 2024. Mechanisms and implications of bacterial–fungal competition for soil resources. ISME J. 18, wrae073. https://doi.org/10.1093/ismejo/wrae073
- Yu, K., van den Hoogen, J., Wang, Z., Averill, C., Routh, D., Smith, G.R., Drenovsky, R.E., Scow, K.M., Mo, F., Waldrop, M.P., Yang, Y., Tang, W., Vries, F.T.D., Bardgett, R.D., Manning, P., Bastida, F., Baer, S.G., Bach, E.M., García, C., Wang, Q., Ma, L., Chen, B., He, X., Teurlincx, S., Heijboer, A., Bradley, J.A., Crowther, T. W., 2022. The biogeography of relative abundance of soil fungi versus bacteria in surface topsoil. Earth Syst. Sci. Data 14, 4339–4350. https://doi.org/10.5194/essd-14-4339-2022
- Zhang, Q., Li, X., Liu, J., Liu, J., Han, L., Wang, X., Liu, H., Xu, M., Yang, G., Ren, C., Han, X., 2023. The contribution of microbial necromass carbon to soil organic carbon in soil aggregates. Appl. Soil Ecol. 190, 104985. https://doi.org/10.1016/j.apsoil.2023.104985
- Zhou, R., Liu, Y., Dungait, J.A., Kumar, A., Wang, J., Tiemann, L.K., Zhang, F., Kuzyakov, Y., Tian, J., 2023. Microbial necromass in cropland soils: A global meta analysis of management effects. Global Change Biol. 29, 1998–2014. https://doi.org/10.1111/gcb.16613
- Zhou, T., Sun, J., Ye, C., Jing, X., Liang, E., Lu, X., Mori, A.S., Meadows, M.E., Peñuelas, J., 2025. Climate change is predicted to reduce global belowground ecosystem multifunctionality. Nat. Commun. 16, 9337. https://doi.org/10.1038/s41467-025-64453-4